# Unlearning Evaluation through Subset Statistical Independence

**Chenhao Zhang**[1,2]**, Muxing Li**[2]**, Feng Liu**[2,4*]**, Weitong Chen**[3]**, Miao Xu**[1,4] *

[1]University of Queensland, [2]University of Melbourne, [3]Adelaide University,
[4]RIKEN Center for Advanced Intelligence Project
{chenhao.zhang, miao.xu}@uq.edu.au, muxing.li@student.unimelb.edu.au
feng.liu1@unimelb.edu.au,t.chen@adelaide.edu.au

## Abstract

Evaluating machine unlearning remains challenging, as existing methods typically require retraining reference models or performing membership inference attacks, both of which rely on prior access to training configuration or supervision labels, making them impractical in realistic scenarios. Motivated by the fact that most unlearning algorithms remove a small, random subset of the training data, we propose a subset-level evaluation framework based on statistical independence. Specifically, we design a tailored use of the Hilbert–Schmidt Independence Criterion to assess whether the model outputs on a given subset exhibit statistical dependence, without requiring model retraining or auxiliary classifiers. Our method provides a simple, standalone evaluation procedure that aligns with unlearning workflows. Extensive experiments demonstrate that our approach reliably distinguishes in-training from out-of-training subsets and clearly differentiates unlearning effectiveness, even when existing evaluations fall short. The codes are available at https://github.com/ChildEden/SDE.

## 1 Introduction

Machine Unlearning aims to remove the influence of specific training data samples from a trained model. This capability is crucial for both adversarial-oriented unlearning, where backdoors or corrupted knowledge introduced during training must be eliminated, and privacy-oriented unlearning, where individuals may request their data be erased under data protection regulations such as the "right to be forgotten" (Garg et al., 2020; Bukaty, 2019). To meet this need, recent works (Nguyen et al., 2022; Xu et al., 2024) have developed various unlearning algorithms to eliminate the influence of a subset of training data that needs to be forgotten, i.e., forgetting data, from a trained model.

One key challenge lies in verifying whether an unlearning process has been successful, especially in realistic deployment settings where retraining from scratch is impractical. Existing unlearning works often assess unlearning effectiveness by comparing metrics such as model utility (e.g., accuracy) (Shen et al., 2024; Fan et al., 2024) and retraining time (Tarun et al., 2024; Zhang et al., 2024) against a retrained model. In these evaluations, the closer the unlearned model resembles the retrained model, the better its unlearning effectiveness. Nevertheless, this evaluation paradigm suffers from a major limitation: it relies on access to a retrained model trained with remaining data only, which defeats the purpose of developing a standalone, verifiably unlearned model.

Membership inference attacks (MIA) are often used to evaluate unlearning by testing whether a specific sample was seen during training. Existing MIAs rely on three main cues: 1) confidence scores, assuming models assign higher confidence to training samples (Salem et al., 2019); 2) loss-based criteria, using the empirical gap in training and held-out loss (Yang et al., 2016); and 3) auxiliary classifiers trained on prediction vectors or hidden representations (Shokri et al., 2017). These methods require access to internal training statistics (e.g., loss distributions, confidence ranges) and often rely on shadow models trained with the same data distribution or hyperparameters. Such assumptions rarely hold in post-hoc unlearning evaluation, where the original training setup or sufficient

---

*Co-correspondence.

data is unavailable (Chundawat et al., 2023), making it infeasible to reconstruct loss baselines or train effective attacker models. Moreover, unlearning methods are typically required to remove a small, random subset of the training data (5%–20%) (Nguyen et al., 2022; Xu et al., 2024), creating two practical challenges: 1) limited data sample or label provide too few supervision to reliably train auxiliary classifiers, and 2) per-sample cues like loss or confidence become statistically weak after unlearning because the subset loses co-adaptation with the remaining data during unlearning. In this context, pursuing accurate general per-sample inference is inefficient and misaligned with the unlearning workflows. Instead, what matters is whether the subset as a whole retains any statistically detectable signal of prior training. We therefore propose a shift from sample-wise MIA to subset-level evaluation, where we test statistical dependence among the model outputs on a candidate forgetting set. Our motivation stems from that training participation induces inter-sample dependencies in the model's internal representations due to shared gradient updates and co-adaptation. In contrast, for data never seen during training, such inter-sample dependency should not arise.

In this work, we propose Split-half Dependence Evaluation (SDE) that evaluates the effectiveness of unlearning by determining whether a given subset is in-training data based on statistical dependence among the model's outputs of the subset. Specifically, we adopt the Hilbert-Schmidt Independence Criterion (HSIC) (Gretton et al., 2005a; 2007), a widely used kernel-based measure well-suited for high-dimensional data, and we novelly propose the split-half dependence test where a subset is split into two halves, and the dependence between their activations is computed via HSIC. Our analysis shows that the split-half dependence test catches the inner subset dependence with a shared sample influence component introduced by the model's training. Unlike existing unlearning evaluations, our method: 1) enables unlearning evaluation without needing a retrained reference model; 2) does not rely on auxiliary classifiers or additional model training, and 3) operates on data subsets rather than individual samples, resulting in a simpler and more robust evaluation that better aligns with the overarching goal of unlearning. Extensive experiments on the retrained models demonstrate that our method can effectively identify the in- and out-of-training subsets. Experiments on existing unlearning methods demonstrate that our method can verify unlearning success even in settings where existing evaluations struggle to provide conclusive evidence.

Due to space constraints, we discuss related works-including unlearning evaluation, MIA, and statistical independence-in Appendix B.

## 2 PRELIMINARIES

Let $\mathcal{D}_{\text{tr}}$ denote the original training dataset, and $\mathcal{D}_{\text{te}}$ denote the test dataset. Let $h$ represent a neural network model. Given an input $x \in \mathcal{X}$, the $h(x)$ is the final layer activation. Deep neural networks may consist of many layers, so we use $h_\ell(x) \in \mathbb{R}^{\text{dim}}$ to denote the activation from the $\ell$-th layer with the dimension of $\text{dim}$. Specifically, we use $h_p(x)$ to denote the activation from the penultimate layer, since it is often used as the extracted feature of the input.

### 2.1 MACHINE UNLEARNING

In the context of machine unlearning, the forgetting data, $\mathcal{D}_{\text{f}} \subset \mathcal{D}_{\text{tr}}$, is the subset of the training data whose influence is intended to be removed. Correspondingly, the remaining data is denoted as $\mathcal{D}_{\text{r}} = \mathcal{D}_{\text{tr}} \setminus \mathcal{D}_{\text{f}}$. Given a deep neural network model $h$ with a specific architecture, we consider three variants of the model in the context of unlearning. The original model $h^{\text{or}}$ is trained on $\mathcal{D}_{\text{tr}}$. The unlearned model $h^{\text{un}}$ is obtained by applying an unlearning algorithm to remove the influence of $\mathcal{D}_{\text{f}}$. The retrained model $h^{\text{re}}$ is a special unlearned model, which is trained on $\mathcal{D}_{\text{r}}$ from scratch, as it is usually used as the gold standard.

### 2.2 HILBERT-SCHMIDT INDEPENDENCE CRITERION (HSIC)

HSIC is a kernel-based statistical measure that quantifies the degree of dependence between two random variables. Given two random variables $X$ and $Y$, $\text{HSIC}(X, Y) = \|C_{XY}\|_{\text{HS}}^2$, where $C_{XY}$ is the cross-covariance operator between the reproducing kernel Hilbert spaces (RKHS) of $X$ and $Y$, and $\| \cdot \|_{\text{HS}}$ denotes the Hilbert-Schmidt norm. HSIC value is non-negative and continuous, providing a meaningful scale: the closer HSIC is to zero, the more independent the two variables are; higher values indicate stronger statistical dependence.

Empirically, the $X$ and $Y$ can be two sets of observations with the same sample size, i.e., $|X| = |Y|$. Given kernel functions defined over the respective domains of the variables, the empirical estimator of HSIC can be expressed as

$$\text{HSIC}(X, Y) = \frac{1}{(n-1)^2} Tr(KHLH),$$

where $K$ and $L$ are the kernel matrices for $X$ and $Y$, respectively, and $H$ is the centering matrix $H = I - \frac{1}{n}\mathbf{1}\mathbf{1}^T$. Following (Gretton et al., 2007), we use the Gaussian RBF kernel as the default choice throughout this paper.

## 3 METHOD

Our main motivation stems from when $h$ is the result of a supervised training algorithm $\mathcal{A}$ and trained on $\mathcal{D}_{\text{tr}}$, i.e., $h = \mathcal{A}(\mathcal{D}_{\text{tr}})$, the learned parameters inherently depend on the training data. To catch such dependence, a naive idea would be to directly compute the dependence between network parameters and training data, e.g., $\text{HSIC}(\mathcal{D}_{\text{tr}}, h)$. However, this is problematic because $h$ has only one observation, as there is usually only one trained network, and it consists of millions of parameters, making it statistically unreliable and computationally prohibitive.

Instead, we consider the dependence among $h(x_i)$'s. Intuitively, we treat the deep neural network $h(\cdot)$ as a complex transformation from the input space to the output space. If $h$ consists of randomly initialized parameters, i.e., a random transformation, then the outputs $h(x_i)$ and $h(x_j)$ for any two samples $x_i, x_j \in \mathcal{D}_{\text{tr}}$ should remain independent. When $h = \mathcal{A}(\mathcal{D}_{\text{tr}})$, $h(x_i)$ implicitly depends on $x_j$ through the learned parameters; hence, the outputs $h(x_i)$ and $h(x_j)$ are no longer independent. In contrast, out-of-training samples (i.e., $\mathcal{D}_{\text{te}}$) are not involved in shaping the parameters of $h$, thus their activations should exhibit weaker statistical dependence. Motivated by this, as well as by the fact that unlearning typically forgets a subset of the training data, we introduce the *Split-half Dependence Evaluation (SDE)*. Given a target subset, we split it into two random halves and measure the dependence between their representations. In later subsections, we empirically validate that under our split-half evaluation, in-training subsets exhibit greater dependence than out-of-training subsets. Appendix A presents an analysis showing that a shared, training-induced influence component yields higher split-half dependence for in-training subsets, which can be reflected by a toy experiment in the Appendix A.4. The overall algorithm is presented in Appendix C.

### 3.1 SPLIT-HALF DEPENDENCE OF A SET OF DATA

Given a target set of data $\mathcal{S}$ and a model $h$, we want to measure the dependence between the activations of samples in $\mathcal{S}$ on $h$. Specifically, we randomly divide $\mathcal{S}$ into two equal sets, $\mathcal{S}_1$ and $\mathcal{S}_2$, and propose the Split-half Dependence $H(\mathcal{S}, h)$ by

$$H(\mathcal{S}, h) = \text{HSIC}(h(\mathcal{S}_1), h(\mathcal{S}_2)), \tag{1}$$

where $\mathcal{S}_1 \cup \mathcal{S}_2 = \mathcal{S}, \mathcal{S}_1 \cap \mathcal{S}_2 = \emptyset$, and $|\mathcal{S}_1| = |\mathcal{S}_2|$. Since HSIC is a statistical metric, we follow the practice of Gretton et al. (2007) to shuffle $\mathcal{S}_2$ 200 times to calculate 200 HSIC values for estimating the $H(\mathcal{S}, h)$ distribution. As motivated earlier, the dependence of in-training data ($\mathcal{S}_{\text{IT}} \subset \mathcal{D}_{\text{tr}}$) activations on a trained model $h^{\text{or}}$ should be significantly higher than that of out-of-training data ($\mathcal{S}_{\text{OOT}} \subset \mathcal{D}_{\text{te}}$), i.e., $H(\mathcal{S}_{\text{IT}}, h) > H(\mathcal{S}_{\text{OOT}}, h)$.

We illustrate this difference by plotting the distributions of $H(S_{\text{IT}}, h)$ and $H(S_{\text{OOT}}, h)$ using both the trained model $h^{or}$ and the randomly initialized model $h^{\text{rand}}$. As shown in Figure 1, the split-half dependence of the in-training subset $S_{\text{IT}}$ under the trained model $h^{\text{or}}$ is significantly higher than that of the out-of-training subset $S_{\text{OOT}}$. In contrast, under the randomly initialized model $h^{\text{rand}}$, the two distributions largely overlap, suggesting no distinguishable dependence signal.

To demonstrate the statistical significance, we further conduct a *one-sided Mann–Whitney U-test* (Mann & Whitney, 1947), under the alternative hypothesis of $H(\mathcal{S}_{\text{IT}}, h) > H(\mathcal{S}_{\text{OOT}}, h)$. This U-test is chosen because it is non-parametric and directly tests our alternative hypothesis $H(S_{\text{IT}}, h) > H(S_{\text{OOT}}, h)$ with established statistical rigor. We reject the null hypothesis and accept this alternative if the $p$-value satisfies $p < 0.01$. As shown in Figure 1, the split-half dependence distributions under the trained model $h^{\text{or}}$ differ substantially, with a $p$-value of $4.73 \times 10^{-11} \ll 0.01$,

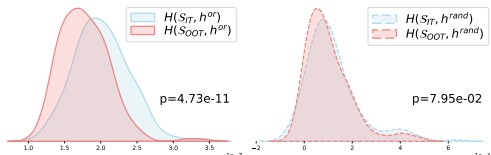 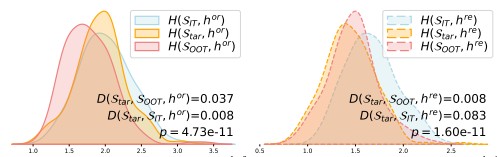

Figure 1: Empirical $H(S, h)$ distributions calculated on in-training subset $\mathcal{S}_{\mathrm{IT}}$ and out-of-training subset $\mathcal{S}_{\mathrm{OOT}}$ using two models: (left) trained model $h^{\mathrm{or}}$ and (right) randomly initialized model $h^{\mathrm{rand}}$.

Figure 2: $H(S, h)$ distributions on the original model $h^{\mathrm{or}}$ (left) and the retrained model $h^{\mathrm{re}}$ (right). The $S_{\mathrm{tar}}$ is in-training for $h^{\mathrm{or}}$ but out-of-training for $h^{\mathrm{re}}$. In the retrained model, the $\mathcal{S}_{\mathrm{tar}}$ becomes significantly closer to the $\mathcal{S}_{\mathrm{OOT}}$ than the $\mathcal{S}_{\mathrm{IT}}$ in terms of HSIC.

indicating that the in-training subsets exhibit significantly higher dependence than out-of-training subsets. In contrast, under the randomly initialized model $h^{\mathrm{rand}}$, the distributions largely overlap, yielding a non-significant $p$-value of 0.0795.

## 3.2 Evaluating the Unlearned Model via $H(\mathcal{S}, h)$

Given the empirical observation that the $H(\mathcal{S}, h)$ distributions diverge significantly between in-training and out-of-training data under a trained model, we exploit this property to assess the efficacy of an unlearning process in forgetting a specific subset from the original training set.

When evaluating an unlearned model $h^{\mathrm{un}}$, we are given a target subset from its forgetting data, i.e., $\mathcal{S}_{\mathrm{tar}} \subseteq \mathcal{D}_{\mathrm{f}}$, and aim to determine whether it resembles in-training or out-of-training data from the unlearned model's perspective. To this end, we reserve a small subset of known in-training data $\mathcal{S}_{\mathrm{IT}} \subset \mathcal{D}_{\mathrm{r}}$ and out-of-training data $\mathcal{S}_{\mathrm{OOT}} \subset \mathcal{D}_{\mathrm{te}}$, referred to as the **reference sets**. In practice, reference sets can be constructed from a small portion of the training and test data that is intentionally retained for auditing or debugging purposes. We first obtain all three $\mathcal{S}$'s split-half dependence distributions, i.e., $H(\mathcal{S}, h^{\mathrm{un}})$, and then compare $H(\mathcal{S}_{\mathrm{tar}}, h^{\mathrm{un}})$ to the reference distributions $H(\mathcal{S}_{\mathrm{IT}}, h^{\mathrm{un}})$ and $H(\mathcal{S}_{\mathrm{OOT}}, h^{\mathrm{un}})$. If $H(\mathcal{S}_{\mathrm{tar}}, h^{\mathrm{un}})$ is significantly closer to $H(\mathcal{S}_{\mathrm{OOT}}, h^{\mathrm{un}})$, we infer that the model exhibits behavior consistent with being trained without $\mathcal{S}_{\mathrm{tar}}$. Conversely, if it is closer to $H(\mathcal{S}_{\mathrm{IT}}, h^{\mathrm{un}})$, we conclude that the $h^{\mathrm{un}}$ is trained with $\mathcal{S}_{\mathrm{tar}}$. For presentation convenience, we define $D(\mathcal{S}_{\mathrm{A}}, \mathcal{S}_{\mathrm{B}}, h)$ to measure the distance between the split-half dependence distributions of two subsets $\mathcal{S}_{\mathrm{A}}$ and $\mathcal{S}_{\mathrm{B}}$ under model $h$. Therefore, an unlearning is considered successful if

$$D(\mathcal{S}_{\mathrm{tar}}, \mathcal{S}_{\mathrm{OOT}}, h^{\mathrm{un}}) < D(\mathcal{S}_{\mathrm{tar}}, \mathcal{S}_{\mathrm{IT}}, h^{\mathrm{un}}), \quad \mathcal{S}_{\mathrm{tar}} \subseteq \mathcal{D}_{\mathrm{f}}. \tag{2}$$

We adopt the Jensen–Shannon Divergence (JSD) (Fuglede & Topsøe, 2004) to compare the split-half dependence distributions due to its favorable properties in our context. Compared to alternatives such as KL divergence and Wasserstein distance, JSD is symmetric, bounded, and numerically stable for empirical distributions with overlapping support. The choice of JSD simplifies the design of an efficient Algorithm 2 and avoids potential instability issues. Specifically, if the JSD is applied,

$$D(\mathcal{S}_{\mathrm{A}}, \mathcal{S}_{\mathrm{B}}, h) := \mathrm{JSD}\big(H(\mathcal{S}_{\mathrm{A}}, h) \,\|\, H(\mathcal{S}_{\mathrm{B}}, h)\big). \tag{3}$$

We conduct experiments with the $h^{\mathrm{or}}$ and the retrained $h^{\mathrm{re}}$. As shown in Figure 2, the distribution of $H(\mathcal{S}_{\mathrm{tar}}, h^{\mathrm{re}})$ is closer to that of the $\mathcal{S}_{\mathrm{OOT}}$ than to the $\mathcal{S}_{\mathrm{IT}}$, yielding $D(\mathcal{S}_{\mathrm{tar}}, \mathcal{S}_{\mathrm{OOT}}, h^{\mathrm{re}}) < D(\mathcal{S}_{\mathrm{tar}}, \mathcal{S}_{\mathrm{IT}}, h^{\mathrm{re}})$. In contrast, when evaluated under the original model $h^{\mathrm{or}}$ trained on the full dataset, the $H(\mathcal{S}_{\mathrm{tar}}, h^{\mathrm{or}})$ remains closer to the $H(\mathcal{S}_{\mathrm{IT}}, h^{\mathrm{or}})$. This supports that Eq. 2 provides a practical signal for detecting whether a group of samples was present in the model's training data.

## 4 Experiment

This section is organized as follows: (1) We begin by conducting controlled experiments on **retrained models**, where we confirm that forgetting data is not involved in the model's training process. This allows us to verify that our proposed method can effectively distinguish between in- and out-of-training subsets, and to investigate its robustness across different conditions from aspects of

model architectures, dataset scales, and representation layers. (2) We compare the proposed statistical independence-based method with commonly used distribution-based metrics. (3) We then apply it to evaluate widely-used unlearning baselines. This enables us to compare their unlearning effectiveness in a unified way. A computational cost analysis and corresponding experiment are included in the Appendix G.

### 4.1 CONTROLLED EXPERIMENTS ON RETRAINED MODELS

We conduct experiments on four benchmark datasets: SVHN (Netzer et al., 2011), CIFAR-10, CIFAR-100 (Krizhevsky et al., 2009), and Tiny-ImageNet. We train two neural network classifiers, the AllCNN (Springenberg et al., 2015) and ResNet-18 (He et al., 2016). The dataset configurations and model architectures are summarized in Appendix D. Appendix E presents a case study applying our method to evaluate a diffusion generative model. For every dataset-model setting, we train 5 retrained models with different random seeds to ensure the reliability of our results. We consider four key aspects: **(1) Kernel bandwidth $\sigma$:** HSIC is kernel-based dependence measure, the kernel bandwidth $\sigma$ plays a crucial role in accurately estimating statistical dependence. **(2) Impact of $|\mathcal{D}_f|$ and $|\mathcal{S}|$:** How does the size of the subset $\mathcal{S}$ and the proportion of forgetting data, i.e., $R = |\mathcal{D}_f|/|\mathcal{D}_{tr}|$, affect the method's ability to detect it? **(3) Layer-level robustness:** Does the method remain valid when evaluating features from internal layers ($h_\ell$) instead of just the last layer? **(4) Training-stage robustness:** Can our method reliably assess unlearning for models at different points in training, e.g., across different epochs? These questions help establish the practical utility and robustness of our method under various realistic scenarios, before applying it to unlearned models.

**Protocol**   We construct a forgetting dataset $\mathcal{D}_f$ by randomly sampling a portion of the original training data $\mathcal{D}_{tr}$, (i.e., $\frac{|\mathcal{D}_f|}{|\mathcal{D}_{tr}|} \in \{5\%, 10\%, 20\%\}$), and use the remaining data $\mathcal{D}_r = \mathcal{D}_{tr} \setminus \mathcal{D}_f$ to train a retrained model $h^{re}$. From both $\mathcal{D}_f$ and $\mathcal{D}_r$, we then sample $n \in \{400, 1000, 2000\}$ instances repeatedly for $m$ times to create $m$ subsets from each. These $2m$ subsets serve as the evaluation targets, with known labels indicating whether each subset originated from the training data of $h^{re}$, i.e., $\{(\mathcal{S}_i, 1)|\mathcal{S}_i \subset \mathcal{D}_r\}$ and $\{(\mathcal{S}_i, 0)|\mathcal{S}_i \subset \mathcal{D}_f\}$. We then apply our method to each target subset $\mathcal{S}_i$ to classify its in- or out-of-training status with Eq. 2. The classification F1 score over the $2m$ subsets indicates how well our method can distinguish between in- and out-of-training data under the $h^{re}$.

#### 4.1.1 KERNEL BANDWIDTH $\sigma$

HSIC adopts the Gaussian kernel by default, defined as:

$$k(x, x') = \exp\left(-\frac{\|x - x'\|^2}{2\sigma^2}\right), \tag{4}$$

where $\sigma$ is the kernel bandwidth. A smaller $\sigma$ results in a more localized kernel that is sensitive to fine-grained differences between samples, which may amplify noise and lead to unstable estimates. In contrast, a larger $\sigma$ produces a smoother kernel, capturing broader structures but leading to higher similarity across all sample pairs.

In this part, we investigate how sensitive our method is to different values of $\sigma$ and aim to identify the effective operating range that yields consistently high performance. We consider two heuristics for selecting $\sigma$: the square root of the activation dimension, i.e., $\sqrt{\dim}$ (Liu et al., 2020), and the widely adopted median heuristic (Schölkopf & Smola, 2002; Garreau et al., 2017) based on pairwise distances between samples. To carry out this analysis, we use the output of $h_p$ and uniformly sample 20 candidate values of $\sigma$ from a continuous range between 1 and $\max(\sqrt{\dim}, \text{Median})$, and evaluate the resulting F1 score under each $\sigma$ value.

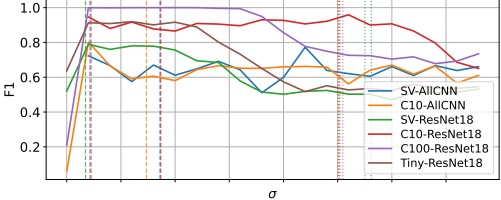

Figure 3: Effect of kernel bandwidth $\sigma$ on F1 score. The **solid lines** show F1 score trends across different $\sigma$ values. Vertical **dashed lines** indicate the heuristic of $\sigma = \sqrt{\dim}$, while **dotted lines** correspond to the median heuristic.

Table 1: F1 score of distinguishing in/out-of-training status of target $\mathcal{S}$, across the amount of $\mathcal{D}_{\mathrm{f}}$ and the size of $\mathcal{S}$.

| $R$ | 5% | | | 10% | | | 20% | | |
|---|---|---|---|---|---|---|---|---|---|
| $|\mathcal{S}|$ | 400 | 1000 | 2000 | 400 | 1000 | 2000 | 400 | 1000 | 2000 |
| SV-AllCNN | 0.62±0.14 | 0.74±0.08 | 0.83±0.09 | 0.50±0.21 | 0.62±0.06 | 0.73±0.12 | 0.72±0.03 | 0.82±0.08 | 0.90±0.06 |
| C10-AllCNN | 0.69±0.07 | 0.79±0.05 | 0.81±0.14 | 0.61±0.05 | 0.66±0.12 | 0.82±0.09 | 0.45±0.00 | 0.71±0.18 | 0.79±0.09 |
| SV-ResNet18 | 0.71±0.09 | 0.91±0.01 | 0.93±0.07 | 0.63±0.10 | 0.78±0.09 | 0.85±0.14 | 0.90±0.03 | 0.96±0.02 | 0.97±0.05 |
| C10-ResNet18 | 0.87±0.06 | 0.97±0.02 | 0.99±0.01 | 0.88±0.04 | 0.95±0.06 | 0.97±0.03 | 0.86±0.10 | 0.96±0.03 | 1.00±0.00 |
| C100-ResNet18 | 0.99±0.01 | 1.00±0.00 | 1.00±0.00 | 0.97±0.05 | 1.00±0.00 | 1.00±0.00 | 0.99±0.01 | 1.00±0.00 | 1.00±0.00 |
| Tiny-ResNet18 | 0.70±0.06 | 0.78±0.06 | 0.92±0.05 | 0.81±0.08 | 0.92±0.03 | 0.98±0.02 | 0.78±0.05 | 0.90±0.06 | 0.98±0.02 |

From Figure 3, we observe that the choice of kernel bandwidth $\sigma$ has a significant impact on the performance of our method. For most dataset-model combinations, the F1 score improves rapidly with increasing $\sigma$ and reaches a plateau within a moderate range. Notably, extremely small $\sigma$ values result in unstable or poor performance, likely due to overly localized kernels that are sensitive to noise. Conversely, excessively large $\sigma$ values may overly smooth the kernel, resulting in loss of discriminative power and a decline in performance. Compared with the median heuristic (dotted lines), the heuristic $\sigma = \sqrt{\dim}$ (dashed lines) generally falls within the high-performing regions, supporting its effectiveness as a practical choice. In the rest of the experiments, unless otherwise specified, we use $\sigma = \sqrt{\dim}$ as the default choice.

### 4.1.2 IMPACT OF $|\mathcal{S}|$ AND $R$

We use the activation of the penultimate layer, denoted as $h_p(x)$, and evaluate models at the checkpoint corresponding to 80% of the total training epochs. We experiment with forgetting ratios $R \in \{5\%, 10\%, 20\%\}$ and target subset sizes $|\mathcal{S}| \in \{400, 1000, 2000\}$.

Table 1 reports the F1 score across various datasets, architectures, forgetting ratios $R$, and test subset sizes $|\mathcal{S}|$. We observe the following consistent trends: 1) **Larger $\mathcal{S}$ improves performance.** Across all settings, the F1 score improves as the target set size increases from 400 to 2000. This is expected, as more samples provide more stable $H(\mathcal{S}, h)$ estimates and reduce variance in distribution comparisons. 2) **Our method remains effective even with small $R$.** While larger forgetting ratios $R$ (e.g., from 5% to 20%) may introduce more changes in the model's output, we observe that our method already achieves competitive performance when $R$ is as small as 5%. This indicates that even subtle representation differences introduced by forgetting a small portion of the data can be detected using our method, demonstrating the sensitivity and robustness of our approach. 3) **Model and dataset matter.** Our method performs particularly well on ResNet-18 architectures (e.g., CIFAR-10 and CIFAR-100), achieving nearly perfect accuracy when $|\mathcal{S}| \geq 1000$ on CIFAR-100. In contrast, performance is lower but still reasonable on AllCNN model.

### 4.1.3 LAYER-WISE GENERALITY

We use the checkpoint at 80% of the total training epochs, set $|\mathcal{S}| = 1000$ and $R = 10\%$. Beyond the final output $h$ and the penultimate activations $h_p$, we further examine intermediate representations to assess the generality of our method across different model layers. For the AllCNN architecture, we select the output of four convolutional layers, $\big[\mathrm{Conv2}, \mathrm{Conv4}, \mathrm{Conv6}, \mathrm{Conv8}\big]$, as intermediate activations. For ResNet-18, we use activations from its four residual blocks, e.g., $\big[\mathrm{Block1}, \mathrm{Block2}, \mathrm{Block3}, \mathrm{Block4}\big]$. We use $\ell \in \{1, 2, 3, 4\}$ to index these intermediate layers, ordered from early (closer to the input) to later layers. We present the dimensionality of activations across layers in Appendix D.

From Figure 4, we observe that our method achieves better performance in deeper layers—such as the penultimate layer $h_p$ and the final output $h$—exhibit higher distinguishability, as they encode more task-specific information. Intermediate layer such as $h_4$ also yields strong signals, particularly on datasets like CIFAR-10 and CIFAR-100 with the ResNet-18 architecture, where the F1 remains above 0.9. This demonstrates that our method is not confined to final-layer outputs but generalizes well across representational levels. As expected, the F1 tends to decrease as we move to lower-level

Table 2: F1 score of SDE across training progress. The performance of retrained models at checkpoints is in Appendix D.

| ckpt at | 10% | 20% | 40% | 80% |
|---|---|---|---|---|
| SV-AllCNN | 0.49±0.02 | 0.51±0.13 | 0.58±0.17 | 0.62±0.06 |
| C10-AllCNN | 0.48±0.15 | 0.66±0.04 | 0.54±0.18 | 0.66±0.12 |
| SV-ResNet18 | 0.58±0.12 | 0.77±0.05 | 0.79±0.05 | 0.78±0.09 |
| C10-ResNet18 | 0.69±0.00 | 0.61±0.13 | 0.72±0.07 | 0.95±0.06 |
| C100-ResNet18 | 0.59±0.06 | 0.60±0.11 | 0.88±0.07 | 1.00±0.00 |
| Tiny-ResNet18 | 0.66±0.22 | 0.86±0.08 | 0.91±0.04 | 0.92±0.03 |

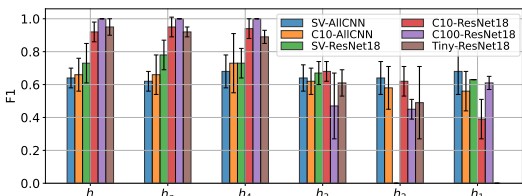

Figure 4: F1 score on activations of different intermediate layers. The x-axis labels from left to right ($h$ to $h_1$) represent layers from output to input. The reason for the missing bar (e.g., the green bar at $h_2$) is further discussed in Section 5.

layers closer to the input (e.g., $h_2$ and $h_1$), especially for shallower networks like AllCNN. This is because earlier layers tend to capture general or low-level features that are less sensitive to the presence or absence of specific samples in training. In summary, our layer-wise evaluation confirms the versatility of the proposed approach, making it suitable for scenarios involving partial model access, transfer learning, or layer-specific unlearning interventions.

### 4.1.4 TRAINING SUFFICIENCY

We investigate whether the training sufficiency of the model affects the effectiveness of our method. In this experiment, we fix the forgetting ratio $R = 10\%$, the target set size $|\mathcal{S}| = 1000$, and use the model's penultimate layer activation $h_p(x)$ for evaluation. We evaluate our method using checkpoints saved at 10%, 20%, 40%, and 80% of the total training epochs, respectively.

From Table 2, the effectiveness of our method improves as the model's training progresses. For example, on SVHN with ResNet18, the F1 increases from 0.58 at 10% training to 0.79 at 40%. On CIFAR10 with ResNet18, the improvement is even more prominent, e.g., from 0.69 to 0.95. This trend suggests that as the model undergoes more training, its internal representations better encode the training data, leading to stronger inter-sample dependencies that our HSIC-based method can capture. Conversely, when the model is under-trained (e.g., at 10%), the representations are not sufficiently informative, making it harder to distinguish between in-training and out-of-training. We also report the performance of retrained models on their respective tasks at these checkpoints in Appendix D Table 7, from where we can observe that signs of overfitting emerge as early as the 20% training stage, even when the overall model performance is still suboptimal. For example, under the CIFAR100-ResNet18 setting, the model achieves only 72.56% training accuracy, yet the test accuracy is only 56.08%, revealing an early onset of generalization gap. This observation is also consistent with a widely acknowledged finding in the MIA literature (Hu et al., 2022): models that are more overfitted to their training data are generally more vulnerable to MIA attacks.

Notably, even at merely 20% of the training process, our method achieves F1 above 0.6 in most cases, indicating early signs of distinguishable dependence. This shows that our method remains applicable even under partially trained models.

### 4.2 COMPARISON WITH DISTRIBUTION DISTANCE METRICS

We compare the proposed statistical independence-based method against two commonly used distribution-based metrics, Maximum Mean Discrepancy (MMD) and Wasserstein distance, to show the advantage of statistical dependence-based methods. We continue to focus on the controlled retrained model and use the CIFAR-10 dataset with the ResNet-18 architecture. The evaluation task remains the same as described in Section 4.1: determining whether a target subset $\mathcal{S}_i$ is part of the training data by comparing $h(\mathcal{S}_i)$ with reference sets $h(\mathcal{S}_{\text{IT}})$ and $h(\mathcal{S}_{\text{OOT}})$. For distribution distance based metrics, $\mathcal{S}_i$ is classified as in-training if $h(\mathcal{S}_i)$ is closer to $h(\mathcal{S}_{\text{IT}})$ than to $h(\mathcal{S}_{\text{OOT}})$, while the opposite assignment applies to out-of-training samples.

According to Table 3, while both MMD and Wasserstein distance exhibit improved performance with larger subset sizes, their accuracy varies significantly with different forgetting ratios and is

Table 3: Comparison with distribution-based metrics

| $R$ | 5% | | | 10% | | | 20% | | |
|---|---|---|---|---|---|---|---|---|---|
| $|\mathcal{S}|$ | 400 | 1000 | 2000 | 400 | 1000 | 2000 | 400 | 1000 | 2000 |
| MMD | 0.63±0.07 | 0.65±0.09 | 0.87±0.12 | 0.45±0.21 | 0.70±0.13 | 0.87±0.14 | 0.63±0.03 | 0.72±0.08 | 0.89±0.11 |
| Wasserstein | 0.70±0.08 | 0.77±0.10 | 0.94±0.11 | 0.52±0.20 | 0.89±0.08 | **0.98±0.03** | 0.72±0.02 | 0.87±0.07 | 0.99±0.02 |
| **SDE (OURS)** | **0.87±0.06** | **0.97±0.02** | **0.99±0.01** | **0.88±0.04** | **0.95±0.06** | 0.97±0.03 | **0.86±0.10** | **0.96±0.03** | **1.00±0.00** |

generally lower than SDE. Notably, SDE consistently achieves higher F1 scores across all settings, even when the subset size is small (e.g., $|\mathcal{S}| = 400$). This suggests that our statistical independence–based approach is more robust than distance-based metrics, particularly when the size of the subset is small. We believe that this advantage stems from SDE directly measuring inter-sample dependency structures, rather than relying on marginal distributional shifts. Moreover, distribution-based methods tend to suffer from higher variance and sensitivity to sample size, which may further degrade their reliability in subset-level evaluation.

## 4.3 EVALUATING UNLEARNING METHODS

All experiments in the previous sections used the retrained model as a controlled object. In this section, we evaluate unlearned models that come from several representative unlearning algorithms. We consider several widely adopted unlearning baselines, including **Random-label** (Fan et al., 2024), **Unroll** (Thudi et al., 2022), **Salun** (Fan et al., 2024), and **Sparsity** (Jia et al., 2023).

**Unlearning Task:** Given an original model $h^{\text{or}}$ trained on the full training dataset $\mathcal{D}_{\text{tr}}$, we simulate sample-wise unlearning by randomly removing a subset of training data with unlearning ratios $R \in \{5\%, 10\%, 20\%\}$.

**Evaluation Protocol:** Given a target subset $\mathcal{S}$ of 1000 samples and reserved reference subsets of the same size, $\mathcal{S}_{\text{IT}} \subset \mathcal{D}_{\text{r}}$ and $\mathcal{S}_{\text{OOT}} \subset \mathcal{D}_{\text{te}}$, the evaluation task is to determine whether $\mathcal{S}$ is in- or out-of-training data. For statistical significance, we sampled $\mathcal{S} \subset \mathcal{D}_{\text{f}}$ for 100 times and report the number of subsets identified as in- or out-of-training. **In this case, an effective unlearned model should have more $\mathcal{S}$ identified as out-of-training and less as in-training, resulting in a higher out-of-training rate (OTR).** To demonstrate that our method is still effective on unlearned models, we sample balanced 100 subsets from $\mathcal{D}_{\text{r}}$ and $\mathcal{D}_{\text{te}}$ as controlled targets and report the F1 score. Noteworthy, the controlled target subsets are not required in practice.

**Other Metrics:** In addition to the OTR, which is based on our proposed method, we follow prior sample-wise unlearning works' experiments (Jia et al., 2023; Fan et al., 2024; Shen et al., 2024) and report other commonly used metrics, including **accuracies** on training sets, as well as membership inference attack success rate (**ASR**). Specifically, $\text{Acc}_{\text{r}}$ and $\text{Acc}_{\text{f}}$ denote the classification task accuracy of the unlearned model on the remaining set $\mathcal{D}_{\text{r}}$ and forgetting set $\mathcal{D}_{\text{f}}$, respectively. These accuracies are used to evaluate the unlearned models' task utility. For the ASR, we also follow existing unlearning works (Fan et al., 2024; Jia et al., 2023) and adopt the prediction confidence-based attack method MIA methods (Song et al., 2019; Yeom et al., 2018). **According to prior works, a desirable unlearning method should have all these metrics that are close to a retrained model.**

**Results:** Table 4 shows the results on CIFAR10-ResNet18 with $R = 10\%$. A complete result with $R \in \{5\%, 10\%, 20\%\}$ is shown in Appendix F. Except for the Sparsity method, our proposed approach consistently achieves high F1 scores (mostly above 0.88), showing its effectiveness in correctly distinguishing whether a given subset $\mathcal{S}$ is in-training or out-of-training for unlearned models. The Retrain model shows very high OTR (87% for $h$ and 94% for $h_p$), indicating that many of the forgetting subsets are indeed identified as out-of-training—a desirable property for a fully unlearned model. From the ASR results, all methods except Sparsity exhibit similar membership inference resistance compared to the Retrain (all around 0.3), making it difficult to judge their unlearning quality solely from ASR. However, the OTR provides a clearer picture: the Random-label method shows strong unlearning effectiveness, with 84% of forgetting subsets no longer recognized as in-training.

Table 4: Evaluating unlearned models on CIFAR10-ResNet18 with $R = 10\%$. $\text{Acc}_r$ and $\text{Acc}_f$ denote unlearned models' training accuracy on the $\mathcal{D}_r$ and $\mathcal{D}_f$, respectively. ASR refers to the success rate of MIA. For these metrics, the closer to the retrained model's the better. For our method, the higher OTR indicates more effective unlearning.

| Method | $\text{Acc}_r$ (%) | $\text{Acc}_f$ (%) | ASR | $h$ | | $h_p$ | |
|---|---|---|---|---|---|---|---|
| | | | | F1 | OTR (%) ↑ | F1 | OTR (%) ↑ |
| Retrain | 98.57±0.08 | 93.25±0.45 | 0.30±0.09 | 0.94±0.03 | 87.00±10.24 | 0.95±0.05 | 94.00±4.94 |
| RandLabel | 98.80±0.04 | 98.63±0.13 | 0.29±0.02 | 0.88±0.12 | **84.00±13.54** | 0.91±0.09 | **83.20±10.11** |
| Unroll | 99.36±0.05 | 99.21±0.11 | 0.30±0.12 | 0.88±0.04 | 3.00±2.19 | 0.90±0.07 | 4.40±4.03 |
| Sparsity | 92.72±0.93 | 90.56±0.82 | 0.42±0.09 | 0.62±0.15 | 50.80±22.61 | 0.59±0.16 | 53.80±24.19 |
| SalUn | 98.66±0.03 | 98.53±0.07 | 0.29±0.02 | 0.85±0.12 | 52.40±21.86 | 0.86±0.15 | 51.80±23.05 |

In contrast, the Unroll method has an extremely low OTR, suggesting that nearly all forgetting subsets are still treated as in-training, indicating ineffective unlearning.

## 5 DISCUSSION AND LIMITATION

**The selection of $\sigma$ and kernel function** Currently, our method relies on kernel-based HSIC to capture statistical dependencies among samples. The choice of the RBF kernel and the bandwidth parameter $\sigma$ directly affects the sensitivity of our metric. As shown in Figure 3 and Figure 6, the choice of kernel bandwidth $\sigma$ critically affects the F1 score. While the heuristic $\sigma = \sqrt{\dim}$ works reasonably well in classification settings, it fails to achieve the best results in diffusion experiments, as seen in Figure 6. This indicates that simple heuristics may not generalize to all scenarios. More adaptive strategies for $\sigma$ selection, or the design of alternative kernel functions tailored to high-dimensional samples, may further improve the robustness and sensitivity of our evaluation approach.

**The selection of reference sets** Our evaluation approach relies on reference sets, which can either be 1) prepared by an authenticated third-party auditor and required to be involved in training or 2) provided by the model owner and archived by the auditor. The choice of reference sets affects the performance of our method. For example, the missing green bar at $h_2$ in Figure 4 occurs because the randomly selected reference sets $\mathcal{S}_{IT}$ and $\mathcal{S}_{OOT}$ fail to satisfy $H(\mathcal{S}_{IT}, h) > H(\mathcal{S}_{OOT}, h)$ under the U-test. Designing strategies for constructing optimal reference sets may be an important direction for improving robustness.

**Advantages of SDE** We view SDE as a step toward practical unlearning evaluation in real deployment scenarios, for several reasons: (1) it requires no retrained reference model; (2) it avoids auxiliary model training and supports flexible cross-layer evaluation; (3) it provides an independence-based perspective rather than distribution-based criteria that can be gamed when directly optimized; and (4) its subset-level focus aligns more closely with existing practical unlearning workflows.

**Rethinking privacy-oriented unlearning evaluation** Our experimental results reveal a critical discrepancy between existing evaluation metrics and our proposed method. For example, in Table 4, existing metrics would suggest that Unroll is effective, as it shows results close to the retrained model. However, our evaluation clearly indicates that Unroll fails to remove the influence of forgetting data, with most forgetting samples still identified as in-training. This discrepancy suggests that relying solely on existing metrics may lead to overestimating the effectiveness of unlearning methods, motivating a rethinking of how unlearning should be rigorously evaluated.

**Unlearning vs. General forgetting** While our approach effectively evaluates unlearning, it may also capture general forgetting phenomena caused by representation drift or catastrophic forgetting. Distinguishing intentional unlearning from natural model degradation is non-trivial and may require incorporating temporal dynamics or additional verification signals.

**Beyond a binary decision** Our current evaluation distinguishes in-/out-of-training by comparing a target subset against a reference set, which does not fully exploit that HSIC is a *continuous*

dependence measure. A promising direction is to *quantify* unlearning by comparing the dependence of the same target subset $\mathcal{S}_{\text{tar}} \subset \mathcal{D}_{\text{f}}$ across different unlearned models. For example, if $H(\mathcal{S}_{\text{tar}}, h^{\text{un1}}) < H(\mathcal{S}_{\text{tar}}, h^{\text{un2}})$, the unlearned model $h^{\text{un1}}$ exhibits better unlearning than $h^{\text{un2}}$.

## 6 CONCLUSION

We propose Split-half Dependence Evaluation (SDE) for evaluating machine unlearning based on statistical dependence among the unlearned model's output representations. By designing a tailored use of the Hilbert–Schmidt Independence Criterion (HSIC), our method enables subset-level evaluation without the need for retrained models or auxiliary classifiers. Our analysis shows the success of SDE is because of a shared influence component that is introduced in training progress. Extensive experiments on classification and diffusion-based generative models demonstrate that our approach reliably identifies the in-training and out-of-training status of small data subsets and provides clear, robust conclusions, even in scenarios where existing evaluations fail to offer decisive evidence.

## ACKNOWLEDGMENTS

CHZ and MX were supported by the Australian Research Council Discovery Project DE230101116. MXL is supported by Australian Research Council (ARC) with the grant number DP230101540. FL is supported by the ARC with the grant number DE240101089, LP240100101, DP230101540 and the NSF&CSIRO Responsible AI program with grant number 2303037.

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

## THE USE OF LARGE LANGUAGE MODELS (LLMS)

We used LLMs solely for polishing the writing of the paper, including grammar correction and phrasing alternatives for clarity and brevity. All content generated with the LLM was treated as suggestions and was reviewed, edited, and verified by the authors, who take full responsibility for the manuscript.

The LLM was *NOT* used in generating technical content; all technical content is author-generated and author-validated.

## A INSIGHTFUL ANALYSIS

In this section, we provide an insight into why the proposed Split-half Dependence Evaluation provably distinguishes subsets drawn from the training data (*in-training*, IT) from those never used in training (*out-of-training*, OOT). Intuitively, training progress leaves a shared "influence footprint" of each example in the learned parameters; this common component appears in both halves of an IT subset, couples their representations, and yields a strictly positive HSIC. In contrast, OOT subsets do not affect the training trajectory and contribute no shared influence; consequently, their two halves remain independent.

### A.1 PRELIMINARIES

**HSIC operator view.** For random variables $X, Y$ with RKHS features $\varphi, \psi$,

$$\text{HSIC}(X, Y) = \|\mathcal{C}_{XY}\|_{\text{HS}}^2, \quad \mathcal{C}_{XY} = \mathbb{E}\big[(\varphi(X) - \mu_X) \otimes (\psi(Y) - \mu_Y)\big].$$

Here $\|\cdot\|_{\text{HS}}$ denotes the Hilbert–Schmidt norm: for an operator $A$ between Hilbert spaces,

$$\|A\|_{\text{HS}}^2 = \sum_i \|Ae_i\|^2,$$

where $\{e_i\}$ is any orthonormal basis of the input space. Equivalently, if $A$ admits a singular value decomposition with singular values $\{\sigma_j\}$, then $\|A\|_{\text{HS}}^2 = \sum_j \sigma_j^2$.

**Local linearization of representations.** Around a reference parameter $\theta_\star$,

$$h_\ell(x; \theta) = h_\ell(x; \theta_\star) + J_\ell(x)\Delta\theta + R_\ell(x), \quad J_\ell(x) = \nabla_\theta h_\ell(x; \theta)\big|_{\theta_\star}, \tag{5}$$

with $\|R_\ell(x)\| = o(\|\Delta\theta\|)$.

**Split-half cross-covariance.** Given a target subset $\mathcal{S}$ with a random split into $\mathcal{S}_1$ and $\mathcal{S}_2$, we define

$$H(\mathcal{S}, h_\ell) = \text{HSIC}\big(h_\ell(\mathcal{S}_1), h_\ell(\mathcal{S}_2)\big).$$

With characteristic kernels, HSIC can be expressed through the RKHS cross-covariance between the two halves. Using the linearization Eq. 5, the dominant term is

$$\mathcal{C}_{\mathcal{S}_1, \mathcal{S}_2} = \mathbb{E}\Big[\big(J_\ell(X)\Delta\theta - \mathbb{E}J_\ell(X)\Delta\theta\big) \otimes \big(J_\ell(X')\Delta\theta - \mathbb{E}J_\ell(X')\Delta\theta\big)\Big], \tag{6}$$

where $X \sim \mathcal{S}_1$ and $X' \sim \mathcal{S}_2$ are independent draws.

**Influence decomposition of parameter shift.** Under ERM with (mini-batch) SGD and mild regularity (Koh & Liang, 2017), the trained parameter admits the approximation

$$\Delta\theta \approx \frac{1}{n} \sum_{x \in \mathcal{D}_{\text{tr}}} \mathcal{I}(x), \qquad \mathcal{I}(x) = -H^{-1}\nabla_\theta \ell(x, \theta_\star), \tag{7}$$

where $H$ is a damped Hessian or a PSD curvature proxy. This holds as a first-order approximation via influence functions.

### A.2 CASE 1: IN-TRAINING SUBSET ($\mathcal{S} \subseteq \mathcal{D}_{\text{tr}}$)

In this case, the influence decomposition Eq. 7 can be refined as

$$\Delta\theta = \underbrace{\frac{1}{n} \sum_{x \in \mathcal{S}} \mathcal{I}(x)}_{\Delta\theta_S} + \underbrace{\frac{1}{n} \sum_{x \in \mathcal{D}_{\text{tr}} \setminus \mathcal{S}} \mathcal{I}(x)}_{\Delta\theta_{\text{rest}}}. \tag{8}$$

Here $\Delta\theta_S$ is the shared component contributed by $\mathcal{S}$ itself. Note that both halves $\mathcal{S}_1$ and $\mathcal{S}_2$ inherit this same $\Delta\theta_S$, which creates correlation across the split.

Starting from the split-half cross-covariance Eq. 6 and the IT decomposition Eq. 8, write

$$\Delta\theta = \Delta\theta_S + \Delta\theta_{\text{rest}}.$$

Abbreviate $A = J_\ell(X)$ and $A' = J_\ell(X')$, and their centered versions $\widetilde{A} = A - \mathbb{E}A$, $\widetilde{A}' = A' - \mathbb{E}A'$. Then Eq. 6 is

$$\begin{aligned}
\mathcal{C}_{\mathcal{S}_1, \mathcal{S}_2} &= \mathbb{E}\Big[(A\Delta\theta - \mathbb{E}A\Delta\theta) \otimes (A'\Delta\theta - \mathbb{E}A'\Delta\theta)\Big] \\
&= \mathbb{E}\Big[(\widetilde{A}\Delta\theta) \otimes (\widetilde{A}'\Delta\theta)\Big] \\
&= \mathbb{E}\Big[(\widetilde{A}(\Delta\theta_S + \Delta\theta_{\text{rest}})) \otimes (\widetilde{A}'(\Delta\theta_S + \Delta\theta_{\text{rest}}))\Big] \\
&= T_{SS} + T_{Sr} + T_{rS} + T_{rr},
\end{aligned} \tag{9}$$

where the four blocks are

$$T_{SS} = \mathbb{E}[(\widetilde{A}\Delta\theta_S) \otimes (\widetilde{A}'\Delta\theta_S)], \quad T_{Sr} = \mathbb{E}[(\widetilde{A}\Delta\theta_S) \otimes (\widetilde{A}'\Delta\theta_{\text{rest}})],$$

$$T_{rS} = \mathbb{E}[(\widetilde{A}\Delta\theta_{\text{rest}}) \otimes (\widetilde{A}'\Delta\theta_S)], \quad T_{rr} = \mathbb{E}[(\widetilde{A}\Delta\theta_{\text{rest}}) \otimes (\widetilde{A}'\Delta\theta_{\text{rest}})].$$

Condition on the fixed split so that $\Delta\theta_S$ and $\Delta\theta_{\text{rest}}$ are constant vectors, while $X \sim \mathcal{S}_1$ and $X' \sim \mathcal{S}_2$ are independent. Then by independence,

$$\mathbb{E}[U(X) \otimes V(X')] = \mathbb{E}[U(X)] \otimes \mathbb{E}[V(X')].$$

Hence

$$T_{Sr} = \mathbb{E}[\widetilde{A}\Delta\theta_S] \otimes \mathbb{E}[\widetilde{A}'\Delta\theta_{\text{rest}}] = (\mathbb{E}[\widetilde{A}]\Delta\theta_S) \otimes (\mathbb{E}[\widetilde{A}']\Delta\theta_{\text{rest}}) = \mathbf{0},$$

and symmetrically $T_{rS} = \mathbf{0}$. Moreover,

$$T_{rr} = \mathbb{E}[\widetilde{A}\Delta\theta_{\text{rest}}] \otimes \mathbb{E}[\widetilde{A}'\Delta\theta_{\text{rest}}] = \mathbf{0},$$

again because $\mathbb{E}[\widetilde{A}] = \mathbf{0}$ and $\mathbb{E}[\widetilde{A}'] = \mathbf{0}$ by centering. Therefore,

$$\mathcal{C}_{\mathcal{S}_1, \mathcal{S}_2} = T_{SS} = \mathbb{E}[(\widetilde{A}\Delta\theta_S) \otimes (\widetilde{A}'\Delta\theta_S)]. \tag{10}$$

Write $\Delta\theta_S$ via the two halves:

$$\Delta\theta_S = \frac{1}{n}(I_{\mathcal{S}_1} + I_{\mathcal{S}_2}), \qquad I_{\mathcal{S}_1} = \sum_{x \in \mathcal{S}_1} \mathcal{I}(x), \ \ I_{\mathcal{S}_2} = \sum_{x \in \mathcal{S}_2} \mathcal{I}(x),$$

which are constant vectors given the split. Then

$$\widetilde{A}\Delta\theta_S = \frac{1}{n}(\widetilde{A}I_{\mathcal{S}_1} + \widetilde{A}I_{\mathcal{S}_2}), \qquad \widetilde{A}'\Delta\theta_S = \frac{1}{n}(\widetilde{A}'I_{\mathcal{S}_1} + \widetilde{A}'I_{\mathcal{S}_2}),$$

and Eq. 10 becomes

$$\mathbb{E}[(\widetilde{A}\Delta\theta_S) \otimes (\widetilde{A}'\Delta\theta_S)] = \frac{1}{n^2} \mathbb{E}\Big[(\widetilde{A}I_{\mathcal{S}_1} + \widetilde{A}I_{\mathcal{S}_2}) \otimes (\widetilde{A}'I_{\mathcal{S}_1} + \widetilde{A}'I_{\mathcal{S}_2})\Big]$$

$$= \frac{1}{n^2} \sum_{a \in \{\mathcal{S}_1, \mathcal{S}_2\}} \sum_{b \in \{\mathcal{S}_1, \mathcal{S}_2\}} \mathbb{E}[(\widetilde{A}I_a) \otimes (\widetilde{A}'I_b)]. \tag{11}$$

Since $I_a, I_b$ are constant and we already work with the centered covariance ($\widetilde{A} = A - \mathbb{E}A$, $\widetilde{A}' = A' - \mathbb{E}A'$), we can have

$$\mathbb{E}[(J_\ell(X)\Delta\theta_S) \otimes (J_\ell(X')\Delta\theta_S)] = \frac{1}{n^2} \sum_{a \in \{\mathcal{S}_1, \mathcal{S}_2\}} \sum_{b \in \{\mathcal{S}_1, \mathcal{S}_2\}} \mathbb{E}[(J_\ell(X)I_a) \otimes (J_\ell(X')I_b)], \tag{12}$$

where $I_{\mathcal{S}_1} = \sum_{x \in \mathcal{S}_1} \mathcal{I}(x)$ and $I_{\mathcal{S}_2} = \sum_{x \in \mathcal{S}_2} \mathcal{I}(x)$ are constant vectors given the split.

Eq. 11 contains four terms: two "same-half" terms ($a = b$) and two "cross-half" terms ($a \neq b$). Conditioned on a fixed split, $X \sim \mathcal{S}_1$ and $X' \sim \mathcal{S}_2$ are independent, hence

$$\mathbb{E}[J_\ell(X) \otimes J_\ell(X')] = \mathbb{E}[J_\ell(X)] \otimes \mathbb{E}[J_\ell(X')].$$

Therefore, after centering, each same-half term cancels:

$$\underbrace{\mathbb{E}[(J_\ell(X)I_{\mathcal{S}_1}) \otimes (J_\ell(X')I_{\mathcal{S}_1})] - \mathbb{E}[J_\ell(X)I_{\mathcal{S}_1}] \otimes \mathbb{E}[J_\ell(X')I_{\mathcal{S}_1}]}_{=0},$$

$$\underbrace{\mathbb{E}[(J_\ell(X)I_{\mathcal{S}_2}) \otimes (J_\ell(X')I_{\mathcal{S}_2})] - \mathbb{E}[J_\ell(X)I_{\mathcal{S}_2}] \otimes \mathbb{E}[J_\ell(X')I_{\mathcal{S}_2}]}_{=0}.$$

Hence only the cross-half contributions remain:

$$\mathbb{E}[(J_\ell(X)I_{\mathcal{S}_1}) \otimes (J_\ell(X')I_{\mathcal{S}_2})] + \mathbb{E}[(J_\ell(X)I_{\mathcal{S}_2}) \otimes (J_\ell(X')I_{\mathcal{S}_1})]. \tag{13}$$

These cross-terms are non-vanishing because the shared component

$$\Delta\theta_S = \frac{1}{n}(I_{\mathcal{S}_1} + I_{\mathcal{S}_2}) \neq 0$$

enters both halves (under non-degenerate $J_\ell$ statistics). Consequently,

$$\mathcal{C}^{\text{IT}}_{\mathcal{S}_1, \mathcal{S}_2} \neq \mathbf{0} \qquad \text{and} \qquad H(S_{\text{IT}}, h_\ell) = \|\mathcal{C}^{\text{IT}}_{\mathcal{S}_1, \mathcal{S}_2}\|^2_{\text{HS}} > 0.$$

**Intuition (IT case).** Even though $\mathcal{S}_1$ and $\mathcal{S}_2$ are disjoint subsets, they are not independent once $S$ has influenced the parameters: both halves share the same "fingerprint" $\Delta\theta_S$ in the model. This induces a dependence across the split and guarantees a strictly positive HSIC.

## A.3 CASE 2: OUT-OF-TRAINING SUBSET ($\mathcal{S} \cap \mathcal{D}_{\mathrm{tr}} = \emptyset$)

When $S$ is completely unseen during training, it does not contribute to $\Delta\theta$. Formally,

$$\Delta\theta_S = 0, \qquad \Delta\theta = \Delta\theta_{\mathrm{rest}}.$$

Substituting into Eq. 6, we obtain

$$\mathcal{C}_{\mathcal{S}_1,\mathcal{S}_2}^{\mathrm{OOT}} = \mathbb{E}\Big[ (J_\ell(X)\Delta\theta_{\mathrm{rest}} - \mathbb{E}J_\ell(X)\Delta\theta_{\mathrm{rest}}) \otimes (J_\ell(X')\Delta\theta_{\mathrm{rest}} - \mathbb{E}J_\ell(X')\Delta\theta_{\mathrm{rest}}) \Big]. \tag{14}$$

By comparing with the Eq. 13 in the IT case, there is no shared $\Delta\theta_S$ and cross-half influence in the OOT case. Factorizing the constant vector $\Delta\theta_{\mathrm{rest}}$, this becomes

$$\Big( \mathbb{E}[J_\ell(X) \otimes J_\ell(X')] - \mathbb{E}[J_\ell(X)] \otimes \mathbb{E}[J_\ell(X')] \Big)(\Delta\theta_{\mathrm{rest}} \otimes \Delta\theta_{\mathrm{rest}}).$$

Since $X \sim \mathcal{S}_1$ and $X' \sim \mathcal{S}_2$ are independent draws from disjoint halves of an OOT set,

$$\mathbb{E}[J_\ell(X) \otimes J_\ell(X')] = \mathbb{E}[J_\ell(X)] \otimes \mathbb{E}[J_\ell(X')], \tag{15}$$

and the entire expression vanishes:

$$\mathcal{C}_{\mathcal{S}_1,\mathcal{S}_2}^{\mathrm{OOT}} = \mathbf{0}, \quad \text{so} \quad H(S_{\mathrm{OOT}}, h_\ell) = 0.$$

**Intuition (OOT case).** OOT subsets never influenced the learned parameters, so they do not inject any shared component across the halves. After conditioning on the split, $\mathcal{S}_1$ and $\mathcal{S}_2$ are independent samples with no common "training footprint".

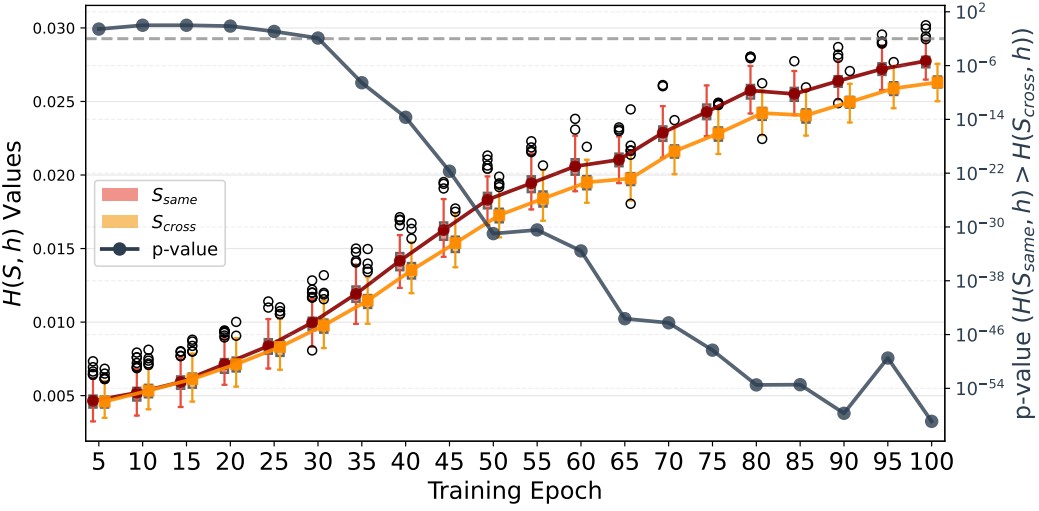

Figure 5: $H(S, h)$ distribution changes along the training epoch. The dashed gray line is the $p = 0.01$ baseline.

## A.4 FIXED BATCH MEMBERSHIP EXPERIMENT

In this subsection, we use a toy experiment to better reflect our main intuition, which is also the basic idea we presented in the above analysis – dependence among outputs arises because training samples co-occur in gradient computations and parameter updates during training.

The toy experiment is based on the following insight: **if the model is trained with fixed (rather than randomly sampled) mini-batches, then samples within the same batch should exhibit significantly stronger dependence compared to samples drawn from different batches.** To validate

this, we constructed a binary classification task with 10,000 data points and 10-dimensional features. We set the batch size to 64 and fixed the batch membership at the beginning of training. A single hidden layer MLP, hidden dimension 128, is used as the model. Under this setup, we expect the dependence within a same-batch subset $S_{\text{same}}$ to be significantly larger than that of a cross-batch subset $S_{\text{cross}}$, that is, $H(S_{\text{same}}, h) > H(S_{\text{cross}}, h)$.

In Figure 5, the gap between the $H(S_{\text{same}}, h)$ and $H(S_{\text{cross}}, h)$ goes larger along with the training steps. The result aligns with this expectation, with a statistical test yielding $p = 2.11 \times 10^{-54} \ll 0.01$, supporting the conclusion that $H(S_{\text{same}}, h) > H(S_{\text{cross}}, h)$.

## B  RELATED WORKS

### B.1  UNLEARNING EVALUATION

Machine unlearning (Bourtoule et al., 2021; Nguyen et al., 2022; Xu et al., 2024) aims to remove the influence of specific training data from machine learning models. It enhances user privacy by complying with regulations such as the "right to be forgotten" (Garg et al., 2020; Bukaty, 2019) and can also mitigate the impact of errors or adversarial contamination in training datasets (Cao & Yang, 2015; Marchant et al., 2022). The effectiveness of an unlearning algorithm depends on its goal and scenario (Kurmanji et al., 2023). When the goal is to remove erroneous or adversarial knowledge (Dong et al., 2025; Jiang et al., 2025; Liu et al., 2022), unlearning effectiveness is often measured by the drop in adversarial attack success rate while maintaining overall model utility. In this paper, we focus on the privacy-oriented unlearning evaluation.

A common evaluation strategy for privacy-oriented unlearning is to use a model retrained from scratch without the forgetting data as the gold standard. Existing approaches assess how similar the unlearned model is to this retrained model in terms of output distributions (Nguyen et al., 2022) or relearning time (Tarun et al., 2024; Zhang et al., 2024). Another intuitive approach leverages Membership Inference Attacks (MIA) to determine whether the forgetting data still leaves a detectable trace in the model. Most prior works still compare the MIA success rate of the unlearned model against that of a retrained model. In contrast, our method enables post-unlearning evaluation without requiring a retrained model. It directly measures whether the influence of the forgotten data persists in the model through statistical dependence analysis, offering a scalable and practical privacy-oriented evaluation approach.

A recent unlearning evaluation study (Tu et al., 2025) also explored the idea of "split sets." In (Tu et al., 2025), their methodological principle is the cryptographic indistinguishability that requires the unlearned model to be computationally indistinguishable from retrained models across adversarially chosen dataset partitions. To test this, they split the dataset multiple times to create different training scenarios for indistinguishability evaluation. In contrast, our methodological principle is statistical independence, which directly tests whether training-induced dependencies persist in the model representations of a given subset. Specifically, we split the given subset into two halves and use the HSIC test to measure the dependence between them. (Tu et al., 2025) requires training multiple models across dataset splits to construct distinguishing games. Ours only conducts statistical dependence testing on the target model's outputs, and no auxiliary model training is needed.

### B.2  MEMBERSHIP INFERENCE ATTACK (MIA)

Membership inference attacks (MIAs) (Hu et al., 2022) aim to determine whether a specific sample was included in a model's training data. MIAs have been widely used as a privacy auditing tool and have strong conceptual alignment with unlearning evaluation, since successful unlearning should erase any membership signal of the forgetting data. A branch of existing MIA techniques (Shokri et al., 2017; Leino & Fredrikson, 2020; Long et al., 2020) trains a binary classifier to identify a data sample's membership regarding the target model's behavior. Another branch of the MIA exploit model confidence (Salem et al., 2019) or loss values (Yang et al., 2016) to identify the membership of a data sample. Existing MIA methods often require additional model training (e.g., shadow models or binary attackers) or access to data labels to compute losses, gradients, or prediction correctness. While these approaches can determine the membership of individual samples, they incur significant overhead and rely on extra assumptions.

In contrast, our method leverages a key property of most unlearning scenarios: forgetting typically targets a subset of the training data rather than isolated samples. By analyzing the statistical dependence within a group of samples, our approach provides a simpler and reliable evaluation. It requires no additional model training, no access to the original training procedure or hyperparameters, and no labeled data, making it a practical tool for post-unlearning privacy assessment.

### B.3 STATISTICALLY SIGNIFICANT DEPENDENCE

Measuring statistical dependence between random variables is a fundamental problem in both statistics (Gretton et al., 2005b) and machine learning (Song et al., 2007). Beyond classical measures such as Pearson correlation and Mutual Information (MI), the Hilbert–Schmidt Independence Criterion (HSIC) (Gretton et al., 2005a; 2007) offers a more general framework, capable of detecting arbitrary dependencies in high-dimensional spaces without requiring explicit density estimation. Our work leverages HSIC to quantify the residual dependence among the representations of a set of samples and build a pipeline for evaluating unlearning effectiveness. If unlearning is effective, the representations of the forgetting data should appear statistically independent, indicating the removal of training influence. By focusing on group-level dependence rather than individual sample behavior, our approach offers a robust and statistically meaningful criterion for post-unlearning evaluation.

## C ALGORITHM FOR UNLEARNING EVALUATION

We decompose the overall unlearning evaluation process into three:

Algorithm 1 (*estimate_hsic_distribution*) first estimates the distribution of HSIC values for a target subset by repeatedly permuting the data. This forms the statistical foundation for detecting whether a subset shows significant dependence.

---

**Algorithm 1** estimate_hsic_distribution() # Section 3.1

---

**Require:**
  Model $h$
  Subset $\mathcal{S}$
  Permutation times $T$
**Ensure:** $H(\mathcal{S}, h)$
  $\mathcal{S}_1 = \mathcal{S}\left[: len(\mathcal{S})//2\right]$
  $\mathcal{S}_2 = \mathcal{S}\left[len(\mathcal{S})//2 :\right]$
  $H = [\,]$
  **for** in $range(T)$ **do**
    $\mathcal{S}_2 = \text{RandomShuffle}(\mathcal{S}_2)$
    $V = HSIC(h(\mathcal{S}_1), h(\mathcal{S}_2))$
    $H.append(V)$
  **end for**
  **return** $H$

---

Algorithm 2 (*is_in_training*) then evaluates whether a candidate subset belongs to the training set. It compares the HSIC distribution of the target set with reference in-training and out-of-training sets using the Jensen–Shannon divergence (JSD), classifying the set as in-training if its HSIC profile is closer to the in-training distribution.

Finally, Algorithm 3 (*unlearn_eval*) quantifies the OOT rate of the unlearned model by sampling multiple subsets from the forgetting data $\mathcal{D}_{\mathrm{f}}$, and checking how many of these subsets are successfully identified as out-of-training. A higher OOT rate indicates more effective unlearning, as more forgetting data are recognized as being removed from the training set.

## D IMPLEMENTATION DETAILS

For the experiments on the classification task, we used mini-batch stochastic gradient descent (SGD) with a weight decay of $5 \times 10^{-4}$. The batch size was set to 256 for SVHN and CIFAR-10, and 128

---

**Algorithm 2** is_in_training() # Section 3.2

---

**Require:**
  Model $h$
  Permutation times $T$
  Target set $\mathcal{S}_{\text{tar}}$
  Reference sets $\mathcal{S}_{\text{IT}}$ and $\mathcal{S}_{\text{OOT}}$
**Ensure:** Is $\mathcal{S}_{\text{tar}}$ in-training data?
  $H(\mathcal{S}_{\text{tar}}, h) = \text{estimate\_hsic\_distribution}(h, H(\mathcal{S}_{\text{tar}}), T)$
  $H(\mathcal{S}_{\text{IT}}, h) = \text{estimate\_hsic\_distribution}(h, H(\mathcal{S}_{\text{IT}}), T)$
  $H(\mathcal{S}_{\text{OOT}}, h) = \text{estimate\_hsic\_distribution}(h, H(\mathcal{S}_{\text{OOT}}), T)$

  $D(\mathcal{S}_{\text{tar}}, \mathcal{S}_{\text{IT}}, h) = JSD(H(\mathcal{S}_{\text{tar}}, h), H(\mathcal{S}_{\text{IT}}, h))$
  $D(\mathcal{S}_{\text{tar}}, \mathcal{S}_{\text{OOT}}, h) = JSD(H(\mathcal{S}_{\text{tar}}, h), H(\mathcal{S}_{\text{OOT}}, h))$
  **return** $D(\mathcal{S}_{\text{tar}}, \mathcal{S}_{\text{IT}}, h) < D(\mathcal{S}_{\text{tar}}, \mathcal{S}_{\text{OOT}}, h)$

---

---

**Algorithm 3** unlearn_eval()

---

**Require:**
  Unlearned model $h^{\text{un}}$
  Permutation times $T$
  Forgetting training set $\mathcal{D}_{\text{f}}$
  Reference sets $\mathcal{S}_{\text{IT}}$ and $\mathcal{S}_{\text{OOT}}$
  Size of target set $n$
  Number of target sets $m$
**Ensure:** OOT rate
  $target\_list = [\,]$
  **for** in range($m$) **do**
    sampling $\mathcal{S}_i \in \mathcal{D}_{\text{f}}$, $|\mathcal{S}_i| = n$
    $target\_list.append(\mathcal{S}_i)$
  **end for**
  OOT_Count=0
  **for** $\mathcal{S}_{\text{tar}}$ in $target\_list$ **do**
    **if not** is_in_training($h^{\text{un}}$,$T$,$\mathcal{S}_{\text{tar}}$,$\mathcal{S}_{\text{IT}}$,$\mathcal{S}_{\text{OOT}}$) **then**
      OOT_Count+=1
    **end if**
  **end for**
  **return** OOT_Count / $m$

---

for CIFAR-100 and Tiny-ImageNet. The initial learning rate was 0.01 for SVHN and 0.1 for all other datasets. All models were trained for the number of epochs listed in Table 5, and the corresponding training, evaluation, and test accuracies are also reported in the table.

Table 5: Training, eval, and test accuracies for different architectures and datasets.

| Dataset-Arch | Ep | TRAIN Acc | EVAL Acc | TEST Acc |
|---|---|---|---|---|
| SV-AllCNN | 20 | 100.00±0.00 | 94.84±0.11 | 94.90±0.05 |
| C10-AllCNN | 50 | 99.41 ±0.02 | 91.95±0.24 | 91.63±0.09 |
| SV-ResNet18 | 20 | 100.00±0.00 | 94.64±0.17 | 94.85±0.10 |
| C10-ResNet18 | 100 | 99.94 ±0.01 | 93.48±0.35 | 93.48±0.22 |
| C100-ResNet18 | 100 | 99.92 ±0.01 | 72.75±0.38 | 73.47±0.21 |
| Tiny-ResNet18 | 100 | 91.10 ±0.23 | 57.20±0.36 | 57.75±0.50 |

Table 6: Activation dimensionalities of different layers.

| | $h_1$ | $h_2$ | $h_3$ | $h_4$ | $h_p$ | $h$ |
|---|---|---|---|---|---|---|
| SVHN-AllCNN | 98304 | 49152 | 12288 | 12288 | 192 | 10 |
| C10-AllCNN | 98304 | 49152 | 12288 | 12288 | 192 | 10 |
| SVHN-ResNet18 | 65536 | 32768 | 16384 | 8192 | 512 | 10 |
| C10-ResNet18 | 65536 | 32768 | 16384 | 8192 | 512 | 10 |
| C100-ResNet18 | 65536 | 32768 | 16384 | 8192 | 512 | 100 |
| Tiny-ResNet18 | 262144 | 131072 | 65536 | 32768 | 512 | 200 |

The Figure 4 in the main paper shows the effectiveness of our method across model layers. Table 6 presents activation dimensionalities of different layers. For each layer activation, we set the $\sigma = \sqrt{\dim}$ by default. Since the dimensionality of the last layer's activation (the logits) of the 10-class tasks is too small, i.e., $\sqrt{\dim} \simeq 3$, we manually set the $\sigma = 128$ for them.

In the Table 2 of the main paper, we explore the impact of training progress on the effectiveness of our method. To demonstrate the models' training sufficiency, we report the model's performance (TrainAcc/TestAcc) at various checkpoints during training in Table 7. We can infer from the table that models exhibit almost no overfitting before 20% of training, but they are not yet fully trained to achieve high performance.

Table 7: Task utilities of models across training progress.

| Data-Arch | Acc | 10% | 20% | 40% | 80% |
|---|---|---|---|---|---|
| SV-AllCNN | Train | 88.09 | 95.73 | 98.51 | 99.92 |
| | Test | 88.16 | 92.97 | 92.54 | 94.26 |
| C10-AllCNN | Train | 92.77 | 99.21 | 100.00 | 100.00 |
| | Test | 90.49 | 92.07 | 94.76 | 94.83 |
| SV-ResNet18 | Train | 77.52 | 84.92 | 89.22 | 98.65 |
| | Test | 71.42 | 78.12 | 80.06 | 91.49 |
| C10-ResNet18 | Train | 76.56 | 87.20 | 91.50 | 99.85 |
| | Test | 71.90 | 80.86 | 85.16 | 93.33 |
| C100-ResNet18 | Train | 58.68 | 72.56 | 81.99 | 99.88 |
| | Test | 52.65 | 56.08 | 57.14 | 73.31 |
| Tiny-ResNet18 | Train | 60.30 | 84.10 | 91.04 | 91.11 |
| | Test | 53.94 | 57.92 | 57.75 | 57.69 |

All the experiments are conducted on one server with NVIDIA RTX A5000 GPU (24GB GDDR6 Memory) and 12th Gen Intel Core i7-12700K CPUs (12 cores and 128GB Memory). The code was implemented in Python 3.12 and CUDA 12.4. The main Python packages' versions are the following: Numpy 2.2.5; Pandas 2.2.3; Pytorch 2.7.0; Torchvision 0.22.0.

# E   CASE STUDY: DIFFUSION MODELS

We further apply our method to generative models, specifically focusing on Elucidated Diffusion Models (EDM) (Karras et al., 2022), one of the most advanced diffusion-based generation frameworks. Experiments are conducted on the CIFAR10, AFHQv2 (Choi et al., 2020), and FFHQ datasets (Karras et al., 2019). For each dataset, we randomly sample 50% of the data to form the forgetting set $\mathcal{D}_f$, and follow the official EDM training configuration[1] to train a retrained model on the corresponding remaining set $\mathcal{D}_r$. The EDM architecture adopts a U-Net (Ronneberger et al., 2015) structure consisting of an encoder and decoder. For our evaluation, we extract the encoder's output as the input of HSIC, which yields a flattened dimension of $dim = 16384$. In computing HSIC, we set the kernel bandwidth $\sigma$ values to span the range from 1 to the full dimensionality.

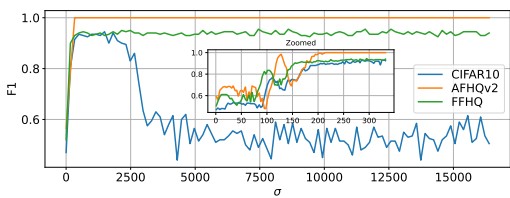

Figure 6: F1 score across $\sigma$ on the diffusion model.

From Figure 6, the F1 score rises sharply as $\sigma$ increases from 1 to roughly 100, after which it stabilizes and maintains high performance across a broad range of $\sigma$. AFHQv2 and FFHQ consistently achieve high F1 scores (above 0.9). In contrast, CIFAR10 exhibits a gradual decline as $\sigma$ continues to increase, mirroring the observation in classification tasks. This is likely due to its lower resolution, which limits the representational richness of the encoder features. The zoomed-in inset highlights the sensitivity in the small-$\sigma$ region, where excessively narrow kernel bandwidths fail to capture meaningful dependencies in the high-dimensional encoder features. Beyond $\sigma \approx 100$, the results

---

[1]https://github.com/NVlabs/edm

stabilize, suggesting that moderate bandwidths are sufficient for reliable HSIC estimation. Following the $\sqrt{\dim}$ heuristic ($\sigma = 128$), the method achieves an F1 score of approximately 0.8 across all datasets, confirming the heuristic as a practical choice for real-world high-dimensional generative models, with further tuning providing additional robustness.

## F   MORE RESULTS ON EVALUATING UNLEARNING METHODS

In the section, we use the CIFAR10-ResNet18 setting as an example and conducted experiments with $R \in \{5\%, 10\%, 20\%\}$. We provide complete results in Table 8.

Table 8: Unlearned models' results on CIFAR10-ResNet18 with $R \in \{5\%, 10\%, 20\%\}$. $\text{Acc}_\text{r}$ and $\text{Acc}_\text{f}$ denote unlearned models' training accuracy on the $\mathcal{D}_\text{r}$ and $\mathcal{D}_\text{f}$, respectively. ASR refers to the success rate of membership inference attacks. For these metrics, the closer to the retrained model's the better. For our method, the higher OTR means that more forgetting data has been identified as out-of-training data, indicating more effective unlearning.

| R | Method | $\text{Acc}_\text{r}$ (%) | $\text{Acc}_\text{f}$ (%) | ASR | $h$ F1 | $h$ OTR (%) ↑ | $h_p$ F1 | $h_p$ OTR (%) ↑ |
|---|--------|----------|----------|-----|----|---------|----|----------|
| | Retrain | 98.70±0.03 | 93.66±0.10 | 0.32±0.12 | 0.85±0.10 | 90.60±10.84 | 0.92±0.06 | 93.20±7.57 |
| 5% | RandLabel | 98.86±0.03 | 98.64±0.13 | 0.25±0.03 | 0.88±0.09 | **79.00±26.26** | 0.90±0.09 | **76.40±28.08** |
| | Unroll | 99.38±0.03 | 99.24±0.22 | 0.28±0.10 | 0.88±0.08 | 0.80±0.75 | 0.88±0.09 | 1.00±2.00 |
| | Sparsity | 93.75±0.54 | 91.52±0.38 | 0.44±0.08 | 0.66±0.29 | 62.00±31.12 | 0.66±0.31 | 62.20±33.07 |
| | SalUn | 98.73±0.02 | 98.54±0.14 | 0.23±0.02 | 0.84±0.11 | 38.80±21.31 | 0.87±0.11 | 37.20±26.61 |
| | Retrain | 98.57±0.08 | 93.25±0.45 | 0.30±0.09 | 0.94±0.03 | 87.00±10.24 | 0.95±0.05 | 94.00±4.94 |
| 10% | RandLabel | 98.80±0.04 | 98.63±0.13 | 0.29±0.02 | 0.88±0.12 | **84.00±13.54** | 0.91±0.09 | **83.20±10.11** |
| | Unroll | 99.36±0.05 | 99.21±0.11 | 0.30±0.12 | 0.88±0.04 | 3.00±2.19 | 0.90±0.07 | 4.40±4.03 |
| | Sparsity | 92.72±0.93 | 90.56±0.82 | 0.42±0.09 | 0.62±0.15 | 50.80±22.61 | 0.59±0.16 | 53.80±24.19 |
| | SalUn | 98.66±0.03 | 98.53±0.07 | 0.29±0.02 | 0.85±0.12 | 52.40±21.86 | 0.86±0.15 | 51.80±23.05 |
| | Retrain | 98.58±0.04 | 92.93±0.27 | 0.25±0.07 | 0.97±0.02 | 99.60±0.49 | 0.98±0.02 | 99.80±0.40 |
| 20% | RandLabel | 98.65±0.06 | 98.64±0.05 | 0.46±0.02 | 0.96±0.03 | 72.60±20.58 | 0.96±0.03 | **64.40±23.27** |
| | Unroll | 99.41±0.04 | 99.27±0.06 | 0.24±0.12 | 0.84±0.20 | 24.40±27.95 | 0.90±0.09 | 7.00±6.72 |
| | Sparsity | 94.28±0.58 | 92.00±0.66 | 0.35±0.07 | 0.75±0.09 | 62.60±18.11 | 0.73±0.09 | 47.40±23.02 |
| | SalUn | 98.51±0.07 | 98.54±0.08 | 0.48±0.03 | 0.93±0.05 | 47.80±25.14 | 0.93±0.05 | 39.60±27.41 |

Across all forgetting ratios, our method consistently achieves high F1 scores (mostly above 0.85), indicating a strong ability to correctly distinguish whether a given subset $\mathcal{S}$ is in-training or out-of-training. Notably, the *Retrain* baseline shows very high OTR (over 87% for $h$ and 93% for $h_p$ at $R = 5\%$), confirming that most forgetting subsets are successfully recognized as out-of-training—an ideal unlearning behavior.

From the ASR perspective, all methods except *Sparsity* achieve membership inference resistance comparable to the retrained model (approximately 0.25–0.32). In this regard, *Unroll* appears to achieve the most effective unlearning, as its ASR is closest to that of the *Retrain* model. However, the OTR metric provides clearer insights:

- **RandLabel** demonstrates strong unlearning effectiveness with high OTR (79%, 84%, 72% for $R = 5\%, 10\%, 20\%$), indicating that a large portion of the forgetting subsets are no longer recognized as in-training.

- **Unroll** consistently yields extremely low OTR (below 5%), suggesting that most forgetting subsets remain in-training, revealing ineffective unlearning despite its high accuracy.

- **Sparsity** achieves moderate OTR (50%–62%), but suffers from low accuracies and higher ASR, showing unstable unlearning quality.

- **SalUn** achieves intermediate OTR performance (38%–52%), indicating partial unlearning but not as effective as *RandLabel* or *Retrain*.

Overall, combining ASR with OTR reveals that *RandLabel* and *Retrain* exhibit the most desirable unlearning behavior, while *Unroll* fails to effectively forget the target subsets.

## G    COMPUTATIONAL COST ANALYSIS

Our proposed SDE consists of the following main computational steps:

1. Network inference. Both the test sample size and the network architecture influence this step's runtime. We exclude this cost from our analysis for two reasons: (1) this is a common step for almost all evaluation methods, including MIAs and ours; (2) in practice, inference only needs to be performed once, and the network outputs can be reused by subsequent steps across different evaluation methods.

2. HSIC calculation. This step requires $O(|S|^2 \times d)$ matrix operations, where $d$ is representation dimension. It can be parallelized efficiently on GPUs. A promising direction for future optimization is to incorporate Nyström kernel approximation to reduce the effective kernel matrix size.

3. Repeatively sample $S_{\text{tar}} \subset \mathcal{D}_{\text{f}}$ for $m$ times for counting OTR. This results $m \times$ above cost.

The overall time complexity could be approximated as $O(m \times |S|^2 \times d)$.

To demonstrate how $m$, $d$, and $|S|$ influence runtime in practice, we conducted a brief experiment under the CIFAR10–ResNet18 setting. The table below reports wall-clock time in seconds. The overall runtime for each entry should include the network inference time corresponding to its subset size. As expected, the runtime increases with the number of repetitions $m$, representation dimension $d$, and subset size $|S|$:

Table 9: Wall-clock time cost (seconds) with different $m$ and $|S|$ when $d = 512$.

|  | Inference Time | $m = 50$ | $m = 100$ | $m = 200$ |
|---|---|---|---|---|
| $|S| = 400$ | 0.29+ | 9.51 | 18.76 | 37.76 |
| $|S| = 1000$ | 0.72+ | 11.06 | 21.92 | 45.85 |
| $|S| = 2000$ | 1.43+ | 29.02 | 57.70 | 116.04 |

Table 10: Wall-clock time cost (seconds) with different $m$ and $|S|$ when $d = 8192$.

|  | Inference Time | $m = 50$ | $m = 100$ | $m = 200$ |
|---|---|---|---|---|
| $|S| = 400$ | 0.29+ | 13.46 | 26.55 | 52.35 |
| $|S| = 1000$ | 0.72+ | 27.15 | 54.18 | 107.99 |
| $|S| = 2000$ | 1.43+ | 84.84 | 168.30 | 335.67 |

