# OpenReview forum: "Unlearning Evaluation through Subset Statistical Independence"
_ICLR.cc/2026/Conference — ICLR 2026 Poster_

### Official Review · Reviewer_QQ14 · 2025-10-31

**Soundness:** 4
**Presentation:** 3
**Contribution:** 4
**Rating:** 8
**Confidence:** 4

**Summary:**

This paper proposes to use Hilbert–Schmidt Independence Criterion to evaluate the effectiveness of unlearning methods. This novel, statistic-based method facilitates evaluation without a retrained reference model or shadow models.

**Strengths:**

1. The proposed method is well-motivated, novel, clear, and effective. This paper is well-presented and easy to read.
2. The research questions studied in 4.1 are important.
3. The discussion in Section 5 is comprehensive and valuable.

**Weaknesses:**

1. Previously, unlearning methods usually played with the metrics, including accuracies and MIA results, via cherry picking hyperparameters with the best performance. It seems HSIC won't stop this game, but just provides one more metric to fit.
2. I think HSIC might be extended for unlearning in generative models, but the authors did not discuss it.

**Questions:**

The authors claim that HSIC evaluation does not require a retrained reference model. But I'm wondering: Is it possible that one day a new unlearning method achieves higher OTR than a retrained model's? If so, how do we analyze this phenomenon?

---

> ### Author Response · Authors · 2025-11-24
> **Response to Reviewer QQ14**
>
> We sincerely thank the reviewer for recognizing the novelty of our work. Below, we provide our responses to the questions raised.
> ## W1. HSIC won't stop this game, but just provides one more metric to fit.
> We sincerely appreciate the reviewer’s recognition of our proposed method as a new metric. While method–metric gaming is a universal challenge for both sides, we would like to respectfully note that our method is inherently more robust against gaming. This is because our method measures statistical dependencies in latent representations, rather than directly optimized training objective such as accuracy or output distributions, which are easier to game.
>
> In addition, we would like to view our method as a step toward unlearning evaluation in real-deployments rather than a replacement of existing method. Since, in many practical unlearning scenarios: (1) no retrained reference model can be easily obtained, and (2) auxiliary attacker models cannot be effectively trained due to limited data or supervision. Under such realistic constraints, our method offers a lightweight and applicable evaluation when existing methods may not be directly usable.
>
> ## W2. Missing discussion of applying our method on generative models.
> Thank you for the suggestion. We have conducted a preliminary exploration of applying our method to generative models. In Appendix E, we present experiments on diffusion generative models (EDM on CIFAR10, AFHQv2, and FFHQ), where our method demonstrates its effectiveness beyond classification tasks. We will add a brief mention of this result in the main paper.
>
> ## Q1. What if OTR exceeding retrained model.
> Thank you for this thought-provoking question. We have not yet observed such cases in our experiments across multiple unlearning methods and settings. Based on our empirical observations, we discuss below why approximate unlearned models' OTR should not exceed the retrained model's, regardless of whether unlearning is incomplete or excessive:
> - If unlearning is incomplete, the forgetting data $\mathcal{D}_f$ retains training-induced dependencies, yielding OTR lower than the retrained model;
> - If unlearning is excessive, aggressive unlearning could disrupt not only training-induced dependencies but also inherent data distribution structures. Such over-unlearning would result indistinguishable statistica dependence between $S_{IT}$ and $S_{OOT}$ yielding OTR lower than the retrained model, and degrade model utility (e.g., accuracy on $\mathcal{D}_r$), which can be detected through standard task performance metrics without requiring a retrained reference.
>
> We hope our clarification addresses the reviewer’s concern. If there are particular settings the reviewer would like us to consider, we are happy to examine them and incorporate additional analysis in the discussion.

---

> ### Author Response · Authors · 2025-11-27
> **Follow-up Response to Reviewer QQ14**
>
> Dear Reviewer QQ14,
>
> With the discussion period closing in less than 7 days, we are writing to kindly follow up on our rebuttal.
>
> We have provided further discussions on the questions you raised, including: **(W1)** disscussed our method in the broader challenge of metric gaming, **(W2)** provided preliminary generative-model results, and **(Q1)** explained why an unlearned model’s OTR should not exceed that of a retrained model.
>
> We sincerely appreciate your recognition of our method’s novelty and effectiveness
>
> We remain available to answer any further questions you might have.
>
> Sincerely,
>
> The authors of  submission 10460

---

### Official Review · Reviewer_3Ap9 · 2025-10-31

**Soundness:** 4
**Presentation:** 4
**Contribution:** 3
**Rating:** 8
**Confidence:** 4

**Summary:**

This paper examines machine unlearning evaluation at the subset level. Motivated by insufficiencies in existing evaluation methods such as retraining (which is computationally expensive) and membership inference attacks (MIA) (which lack effectiveness), the authors propose a tailored metric based on the Hilbert-Schmidt Independence Criterion (HSIC) to measure statistical dependence between model representations and the unlearned subset with a proxy. Experiments demonstrate that the proposed method serves as a reliable evaluation tool for machine unlearning.

**Strengths:**

1. I find the paper very well written and pleasant to read. It is sufficiently motivated by an important problem in machine unlearning and proposes an elegant solution. The presentation is clear and easy to follow.

2. The adoption of subset-level statistical dependence as an evaluation metric is both interesting and clever. It provides a well-defined tool for assessing model-data interactions in machine unlearning (and potentially for measuring neural network memorization more broadly). I believe the proposed method has strong potential to become a standard evaluation tool in the unlearning literature.

3. The paper potentially opens many promising future research directions. I would be particularly interested to see its adaptation to other foundation models, such as those based on contrastive learning methods and generative models.

**Weaknesses:**

1. It appears that the size of the unlearning subset plays an important role in the effectiveness of the proposed method. The experiments demonstrate that SDE works well for unlearning sets comprising 5-20% of the training data, which represents a relatively large proportion. However, consider a scenario where a user requests the unlearning of only a few samples: would SDE remain effective in this case? In other words, is it possible to quantify the minimum subset size at which SDE provides a reliable evaluation?

2. The experiments show that Unroll obtains very low OTR. Is there any explanation for this phenomenon? Moreover, I am curious whether it is possible to attribute unlearning effectiveness at the individual sample level within the subset. While this may seem overly ambitious from a statistical perspective, I wonder if the authors have any insights or preliminary thoughts on this direction.

Note that neither question necessarily requires additional experiments. The authors are welcome to include a discussion of these points and propose reasonable directions for future work.

**Questions:**

See above.

---

> ### Author Response · Authors · 2025-11-24
> **Response to Reviewer 3Ap9**
>
> We sincerely thank the reviewer for recognizing our work as “well written, interesting, and clever, with the potential to open new research directions.” We address the reviewer’s comments point by point below.
>
> ## W1. Lower-bound of the detectable subset size.
> Since our proposed method uses HSIC (Gretton et al., 2007) as the statistical dependence measurement, we refer to the results in their paper where reliable independence testing requires at least 128 samples per variable to achieve sufficient statistical power. Adapting this to our split-half statistic dependence evaluation framework means $|S_1|=|S_2|=128$, which translates to $|S|=256$ for one subset.
>
> Therefore, we conducted additional experiments using the CIFAR10-ResNet18 setting with $|\mathcal{D}_f|=256$, which pushes the forgetting ratio to $R\approx 0.5\%$. We evaluate at $|S|=256$. The result shows that our proposed method achieves an F1 score of 0.79±0.1, which we consider acceptable compared to the F1 score of 0.87 obtained at $R=5\%$ with $|S|=400$ (Table 1). This demonstrates that our proposed method maintains reliable discriminative performance even when the forgetting ratio is as small as $R\approx 0.5\%$ with $|S|=256$.
>
> ## W2.1. Reason of low OTR of Unroll.
> Thank you for pointing this out. The low OTR of Unroll reveals an important insight: while Unroll achieves competitive task performance and ASR scores similar to the retrained model, our proposed method detects that the statistical dependencies among forgetting data representations have not been effectively removed. This suggests that simply reversing gradient updates (like Unroll) may not fully eliminate the training-induced statistical dependencies that our proposed method captures. In contrast, methods like Random-label (OTR: 72-84%) that introduce random labels appear to more effectively break these dependencies, while SalUn, which selectively updates partial parameters (OTR: 38-52%), shows intermediate effectiveness. Although the specific mechanisms behind why certain methods preserve these dependencies while others do not warrant further investigation in future work, our proposed method provides a new perspective to capture whether training-induced dependencies have been truly removed.
>
> ## W2.2. Individual sample level test within the subset.
> We agree this is a fascinating and really challenging direction for statistic-based test. Given our current proposal, one potential approach could involve leave-one-out perturbation analysis: comparing $H(S_{in}, h)$ v.s. $H(S_{in} \cup \{x_i\}, h)$ and, intuitionally, the dependence is expected to reduce if the included $x_i$ is an out-of-training sample. However, this approach faces a fundamental statistical challenge: the perturbation caused by a single sample may be too small to distinguish from statistical noise. This creates a trade-off of while a smaller $|S|$ would amplify the relative impact of individual samples, it would simultaneously reduce the overall statistical power of our proposed method. We appreciate this insightful question and acknowledge that it points to valuable future work.

---

> ### Author Response · Authors · 2025-11-27
> **Follow-up Response to Reviewer 3Ap9**
>
> Dear Reviewer 3Ap9,
>
> With the discussion period closing in less than 7 days, we are writing to kindly follow up on our rebuttal.
>
> We have addressed all of your insightful questions, including: **(W1)** discussed the lower bound of detectable subset size, **(W2.1)** explained the low OTR of Unroll, and **(W2.2)** discussed a potential direction for individual-sample–level evaluation.
>
> We greatly appreciate your positive assessment of our work and hope the added analyses further strengthen clarity.
>
> We remain available to answer any further questions you might have.
>
> Sincerely,
>
> The authors of  submission 10460

---

> > ### Comment · Reviewer_3Ap9 · 2025-11-27
> >
> > Thank you for the detailed rebuttal. As my concerns have been addressed, I maintain my rating.

---

### Official Review · Reviewer_UFX4 · 2025-11-01

**Soundness:** 2
**Presentation:** 2
**Contribution:** 2
**Rating:** 2
**Confidence:** 4

**Summary:**

This paper introduces *Split-half Dependence Evaluation (SDE)*, a new way to measure how well a model has “forgotten” data during machine unlearning. Instead of retraining models or using membership attacks, it checks the statistical independence of model outputs on data subsets using the Hilbert–Schmidt Independence Criterion (HSIC).

**Strengths:**

1. Clear presentation and good illustration.
2. Conceptually new evaluation framework.

**Weaknesses:**

1. Unclear motivation: The benefits of the proposed method should be better explained and carefully compared with other proposed methods. For a proposal of an "evaluation" method, it is especially important to measure/prove/validate its properties, such as robustness, consistency, among others.
2. Unjustified design choice: Some of the design choices are not well-explained, e.g., HSIC with specific RBF kernels, Mann-Whitney U-test, Jensen-Shannon Divergence, among others. To me, it feels artificial to see these components being introduced to the method without a clear justification.

**Questions:**

1. Missing reference: Recently, there have been many related works focusing on unlearning evaluation, e.g., [1], where the idea of split sets is also explored.
2. Weakness 1: Can you explain or provide additional experiments that demonstrate some desirable properties for the proposed evaluation metric, compared to others? Also, can you explain why your evaluation metric is better compared to the existing literature?
3. Weakness 2: Can you provide some brief justification for each of the mentioned components, besides your main conceptual novel proposal (SDE)?

[1]: Tu, Yiwen, Pingbang Hu, and Jiaqi Ma. Towards Reliable Empirical Machine Unlearning Evaluation: A Cryptographic Game Perspective.

---

> ### Author Response · Authors · 2025-11-24
> **Response to Reviewer UFX4 (1)**
>
> We are deeply grateful for the reviewer’s constructive and insightful suggestions, which have helped us improve the clarity and quality of our work. We provide point-by-point responses to the reviewer’s questions below.
>
> ## W1 \& Q2. Benefits, properties, and motivation of the proposed metric.
> ### Our motivation
> Developing an unlearning evaluation method that avoids relying on retraining reference models or training auxiliary attacker models, both of which are costly and often impractical.
> ### Why better than existing evaluation? Benefits.
> 1. **No retrained reference models required.** Our proposal eliminates the need for expensive model retraining and enables evaluation in scenarios where retraining is prohibited or infeasible.
> 2. **No auxiliary model training required, with flexible specific layer evaluation:**
> MIA-based unlearning evaluations typically require training shadow models and attacker models, which necessitate substantial data from the training data distribution. In realistic post-hoc evaluation scenarios, acquiring sufficient data to train effective auxiliary models may be infeasible. Each attacker model targets a specific signal (logits or particular layer outputs), requiring multiple attacker models for multi-layer evaluation. Our method needs no auxiliary models and flexibly evaluates outputs from any layer. It potentially enables testing statistical independence cross outputs at different layers, e.g., $HSIC(S_{1,l_1},S_{1,l_2})$, which is a promising direction for future work.
> ### Properties
> We validate following key properties essential for a robust unlearning evaluation:
> - **Layer-wise robustness,** which ensures the complete removal of forgetting data influence across all model layers, preventing residual effects that may persist in parts of the model during partial unlearning. Figure 4 evaluates 6 different layers across 6 dataset-network settings, showing consistently strong results and graceful improvement toward later layers. Enabling finer layer-wise evaluation.
> - **Robustness to forgetting ratio and evaluation sample budget**, which ensures reliable evaluation even in small-scale forgetting scenarios, where the forgetting data only a small fraction of the training set. Table 1 reports results across nine settings. Our method remains effective in the most challenging case, where only 5% of the data is to be forgotten and the evaluation relies on just 400 samples (less than 1% of the training set).
>
> Compared with distribution-based MMD and Wasserstein distance in distinguishing in-/out-of-training subset, as shown in Section 4.2 Table 3, our proposal is more robust when the forgetting signal is subtle and the evaluation subset is limited.
>
>
> Compared with MIA-based evaluation, our proposal offers the following properties that advances unlearning evaluation:
> 1. Self-contained evaluation capability. It requires no auxiliary model training and operates directly on model representations, enabling practical evaluation in scenarios where training effective auxiliary models is infeasible.
> 2. Cross-layer flexibility without retraining. Our proposal directly computes HSIC on the activations of any layer, eliminating the need to retrain separate attacker models for different layers.

---

> > ### Author Response · Authors · 2025-11-24
> > **Response to Reviewer UFX4 (2)**
> >
> > ## W2 \& Q3. Unjustified design choice.
> > We really appreciate this feedback and the raised concern. We first provide justifications for each choice of "HSIC with specific RBF kernels, Mann-Whitney U-test, Jensen-Shannon Divergence" and then highlight the main design of our proposal.
> > 1. **HSIC with RBF kernel:** HSIC is a well-established kernel-based statistical dependence measure with rigorous theoretical foundations. It has been proven that HSIC (with a characteristic kernel) is equal to 0 if and only if two random variables are independent [3]. To ensure this good property of HSIC, researchers will choose the RBF (Gaussian) kernel, a characteristic kernel, to compute the HSIC value. Another advantage of HSIC is that it can be consistently estimated by U-Statistics or V-Statistics, ensuring that we can always approach its true value if we have a lot of observations. Because of these good properties, HSIC has been used in many research topics, such as model-inversion attack [1], self-supervised learning [2], and independent testing [3].
> >
> > 3. **Mann-Whitney U-test:** The Mann-Whitney U-test is chosen because it is non-parametric and directly tests our alternative hypothesis $H(S_{IT}, h) > H(S_{OOT}, h)$ with established statistical rigor. We introduce the U-test in the paper for ensuring that observed differences are statistically meaningful rather than artifacts of random variation.
> >
> > 3. **Jensen-Shannon Divergence (JSD):** JSD is chosen for comparing the distributions of HSIC values in Eqs. 2–3 for two main reasons. **(1)** Unlike KLD, JSD is symmetric, allowing us to disregard the ordering of the two variables; this simplifies the design of Algorithm 2 and avoids potential instability issues. **(2)** Compared with the Wasserstein distance, JSD is significantly more computationally efficient as it can be computed directly from empirical distributions in $O(n)$, whereas even the Sinkhorn-regularized approximation of the Wasserstein distance still incurs $O(n^2)$ complexity per iteration.
> >
> > Beyond these technical justifications, we highlight that these components contribute to the main design of the proposed Split-half Dependance Evaluation framework:
> > 1. Quantify dependence (HSIC with RBF kernel) and establish significance (Repeated shuffling): Measure how dependent the two halves of a subset are;
> > 2. Compare against references (JSD): Assess whether the target subset's dependence profile resembles in-training or out-of-training data.
> >
> > We thank the reviewer for encouraging us to make these specific choices more clear and will add these justifications to the revised version.
> >
> > ## Q1. Missing reference Tu et al. where the idea of split sets is also explored.
> > We sincerely thank the reviewer for bringing this relevant work to our attention. After carefully reading Tu et al., we are delighted to involve it in our paper's discussion. We clarify the differences between the "split sets" in the two works by highlighting two fundamental distinctions between our approach and that of Tu et al.
> >
> > 1. **Different methodological principles:**
> > - Tu et al.: Cryptographic indistinguishability, which requires the unlearned model to be computationally indistinguishable from retrained models across adversarially chosen dataset partitions. To test this, they split the dataset multiple times to create different training scenarios for indistinguishability evaluation.
> > - Ours: Statistical independence, which directly tests whether training-induced dependencies persist in the model representations of a given subset. Specifically, we split the given subset into two halves and use the HSIC test to measure the dependence between them.
> > 2. **Different practical requirements:**
> > - Tu et al.: Requires training multiple models across dataset splits to construct distinguishing games. In experiments, the raw public dataset is "split into halves", one for training the target model ("target dataset"), and the other for training shadow models for some MIAs.
> > - Ours: Only conducting statistical dependence testing on the target model's outputs and no auxiliary model training needed.
> >
> > ## References
> > [1] Peng, X., Liu, F., Zhang, J., Lan, L., Ye, J., Liu, T., & Han, B. "Bilateral dependency optimization: Defending against model-inversion attacks." SIGKDD 2022.
> >
> > [2] Li, Y., Pogodin, R., Sutherland, D. J., & Gretton, A. "Self-supervised learning with kernel dependence maximization." NeurIPS 2021.
> >
> > [3] Gretton, A., Fukumizu, K., Teo, C., Song, L., Schölkopf, B., & Smola, A. "A kernel statistical test of independence." NeurIPS 2007.

---

> ### Author Response · Authors · 2025-11-27
> **Follow-up Response to Reviewer UFX4**
>
> Dear Reviewer UFX4,
>
> With the discussion period closing in less than 7 days, we are writing to kindly follow up on our rebuttal.
>
> We have carefully addressed all of the questions you raised, including: **(W1&Q2)** clarified the motivation, properties, and benefits of our method, **(W2&Q3)** justified each design choice in our method, and **(Q1)** explained how our use of split subsets fundamentally differs from the Tu et al. cryptographic-game framework.
>
> We would greatly appreciate it if you could take a moment to review our response. If you find that these new results address your concerns, we would value your reconsideration of your assessment.
>
> We remain available to answer any further questions you might have.
>
> Sincerely,
>
> The authors of  submission 10460

---

### Official Review · Reviewer_kGt2 · 2025-11-03

**Soundness:** 3
**Presentation:** 3
**Contribution:** 3
**Rating:** 4
**Confidence:** 3

**Summary:**

This paper proposes a new metric for evaluating unlearning methods, leveraging a clever connection to statistical independence. The main idea behind the paper is to measure the statistical dependence (using a method called the Hilbert-Schmidt Independence Criterion) between different subsets of examples, and use the computed (in)dependence scores as a proxy to evaluate unlearning efficacy. This allows one to evaluate unlearning without needing ground-truth retrained models.

**Strengths:**

- The paper identifies an interesting and important problem (improving the efficiency of unlearning evaluations)
- The idea to use statistical dependence testing as a proxy for retraining is quite interesting and novel. It also seems quite convenient to implement, and thus practical to run in actual unlearning setups.
- The paper conducts a thorough ablation study to understand the effect of different design choices on the efficacy of their approach

**Weaknesses:**

- The theoretical analysis seems quite handwavy and makes lots of approximations without really justifying why we might expect them to be true. To me this is the main concern with the paper, and one that permeates throughout the rest of the weaknesses (see also Q1).
- In Table 4, it's seems concerning that the retrained model does not get 100% according to the metric - doesn't this suggest that the metric is either overly sensitive or otherwise misspecified with respect to the unlearning objective?
- The fact that the method hinges on the selection of good reference sets also limits the practical applicability of the algorithm (this limitation is explicitly acknowledged by the authors).

**Questions:**

- Is there a simple theoretical setting where this evaluation is exactly the right thing to do?
- Did the authors try other kernel functions outside of RBF?
- Can the authors provide a computational cost analysis of their method? What are the main computational steps, and how does the cost scale with the various dataset sizes?

---

> ### Author Response · Authors · 2025-11-24
> **Response to Reviewer kGt2**
>
> We sincerely appreciate the reviewer’s careful and detailed review, and we are grateful for the constructive suggestions. We provide our responses to the reviewer’s questions below.
>
> ## W1 & Q1: A simple theoretical setting where this evaluation is exactly the right thing to do.
> Thank you for raising this important point. To address the Q1, we have added a simple theoretical setting in **Linear regression with full-batch gradient descent**.
>
> Consider a linear model $h_\theta(x) = \theta^\top x$ trained on $\mathcal{D}_{\text{tr}}$ with squared loss. In this convex setting:
> - **Exact parameter decomposition**: The parameter shift admits the exact form $\Delta\theta = \sum_{x \in \mathcal{D}\_{\text{tr}}} I(x)$, where $I(x) = -H^{-1}\nabla_\theta \ell(x)$.
> - **Exact linearity**: Since $h_\theta(x) = \theta^\top x$, the representation is exactly linear in $\Delta\theta$.
> - **Exact HSIC analysis**: For any subset $S \subseteq \mathcal{D}\_{\text{tr}}$, both halves $S_1, S_2$ share the component $\Delta\theta_S = \sum_{x \in S} I(x)$, which induces **strictly positive** split-half dependence, e.g., $\text{HSIC}(h(S_1), h(S_2)) > 0$.
> - **Separation**: For out-of-training subsets $S \cap \mathcal{D}_{\text{tr}} = \varnothing$, we have $\Delta\theta_S = 0$, yielding $\text{HSIC}(h(S_1), h(S_2)) = 0$.
>
> In this fully analyzable convex regime, our proposal achieves exact separation between in-training and out-of-training subsets.
>
>
> ## W2. Retrained model does not get 100% OTR.
> We thank the reviewer for raising this question. We would like to clarify that, in practice, an OTR below 100% on retrained data is under expectation.
>
> This is because deep networks generalize across samples. Even after full retraining, deleted samples retain structural similarities to the remaining training data due to shared class-level features, inductive biases, and optimization dynamics. These factors naturally induce some residual dependence in their representations, so an OTR below 100% agrees with standard behavior in modern deep models.
>
>
> ## Q2. Kernel functions other than RBF
> HSIC is a well-established kernel-based statistical dependence measure with rigorous theoretical foundations. It has been proven that HSIC (with a characteristic kernel) is equal to 0 if and only if two random variables are independent [3]. To ensure this good property of HSIC, researchers widely choose the RBF (Gaussian) kernel [1,2,3], a characteristic kernel, to compute the HSIC value. Given RBF's theoretical soundness and strong empirical performance, it serves as a principled default choice for our method.
>
> ## Q3. Computational cost analysis.
> We greatly appreciate this valuable suggestion and have incorporated the corresponding analysis into the revised manuscript. Our proposal's overall complexity approximated as $O(m \times |S|^{2} \times d)$, which consists of the following computational steps:
>
> 1. **Network inference.** The runtime of this step depends on both the test sample size and the network architecture. We exclude this cost from our analysis for two reasons: (1) it is a common step shared by almost all evaluation methods, including MIAs and ours; and (2) in practice, inference needs to be performed only once, and the resulting network outputs can be reused by subsequent steps across different evaluation methods.
> 2. **HSIC computation.** This step requires $O(|S|^{2} \times d)$ matrix operations, where $|S|$ is the size of given subset and $d$ is the representation dimension. It can be efficiently parallelized on GPUs. A promising direction for further optimization is to incorporate Nyström kernel approximation to reduce the effective kernel matrix size.
> 3. **Repeated sampling of $S_{\text{tar}} \subset \mathcal{D}_f$ for $m$ rounds** to count OTR. This introduces a multiplicative factor of $m$ to the above cost.
>
> We highlight that the dominant factors affecting runtime are $m$, $d$, and $|S|$. In the revised manuscript, we have added a time cost analysis section (Appendix G), which empirically shows that the runtime increases consistently with the number of repetitions $m$, the representation dimension $d$, and the subset size $|S|$.
>
>
> ## References
> [1] Peng, X., Liu, F., Zhang, J., Lan, L., Ye, J., Liu, T., & Han, B. "Bilateral dependency optimization: Defending against model-inversion attacks." SIGKDD 2022.
>
> [2] Li, Y., Pogodin, R., Sutherland, D. J., & Gretton, A. "Self-supervised learning with kernel dependence maximization." NeurIPS 2021.
>
> [3] Gretton, A., Fukumizu, K., Teo, C., Song, L., Schölkopf, B., & Smola, A. "A kernel statistical test of independence." NeurIPS 2007.

---

> ### Author Response · Authors · 2025-11-27
> **Follow-up Response to Reviewer kGt2**
>
> Dear Reviewer kGt2,
>
> With the discussion period closing in less than 7 days, we are writing to kindly follow up on our earlier rebuttal.
>
> We have provided detailed responses to your concerns and questions, including: **(W1\&Q1)** provided a simple theoretical setting, **(W2)** clarified why retrained models do not reach 100% OTR, **(Q2)** explained our choice of the RBF kernel, and **(Q3)** provided a detailed computational cost analysis.
>
> We would greatly appreciate it if you could take a moment to review our response. If the additional analyses address your concerns, we would be grateful for your reconsideration.
>
> We remain available for any further questions you may have.
>
> Sincerely,
>
> The Authors of Submission 10460

---

### Meta-Review · Area_Chair_wVyz · 2026-01-04

**Summary:**

The reviewers raised several concerns, and most of them were addressed by the authors. One of the most important issues, raised by Reviewer kGt2, was the lack of a clean theoretical setting where the proposed evaluation is provably correct. This was partially addressed in the rebuttal by adding a simple linear regression example with full-batch gradient descent, where the split-half HSIC test exactly separates in-training and out-of-training subsets. While this does not fully explain deep networks, it provides a sanity check and strengthens the theoretical basis.

Other concerns focused on justification of design choices (HSIC with RBF kernel, Mann-Whitney U-test, Jensen-Shannon divergence) and computational cost. These were adequately clarified in the rebuttal using standard arguments from the literature, along with an explicit complexity analysis. Questions about why a retrained model does not achieve 100% OTR were addressed by explaining that residual dependence is expected due to generalization in deep representations, and that OTR should be interpreted as a behavioral similarity measure rather than a strict gold standard.

More critical feedback from Reviewer UFX4 centered on motivation and framing-specifically, why this evaluation metric is preferable and what properties it satisfies. While this concern is only partially resolved and reflects a presentation issue, it does not challenge the novelty or empirical validity of the method. Other reviewers’ concerns (e.g., minimum subset size, behavior of Unroll, applicability beyond classification) were addressed with additional analysis or discussion, and at least one reviewer explicitly confirmed their concerns were resolved.

Overall, the rebuttal resolves most of the major technical questions. Given the novelty and empirical support for the proposed evaluation framework, I suggest its acceptance.

**Reviewer Concerns:**

Reviewer kGt2's main concerns were largely addressed, especially the requests for (1) a clean theoretical setting where the method is exactly correct, (2) justification of the kernel choice, and (3) a computational cost analysis. The rebuttal's linear regression plus full-batch gradient descent example directly answers the "simple setting" question and meaningfully strengthens the paper's theoretical grounding. The added complexity breakdown also resolves the request for runtime scaling. The explanation for using an RBF kernel for HSIC is standard and sufficient. Two points are only partially resolved: the reviewer's broader worry that the deep-network theory is still approximation-heavy (linearization and influence-style arguments remain heuristic outside the given example), and the "retrain not 100% OTR" concern, which the rebuttal explains plausibly via generalization-induced residual dependence but still benefits from clearer framing in the paper that OTR is a behavioral similarity score rather than an absolute gold-standard target.

Reviewer UFX4's concerns about "ad-hoc" design choices were addressed by the rebuttal: the authors provide reasonable justifications for HSIC with an RBF kernel, and Jensen-Shannon divergence for stable, symmetric distribution comparison. The missing-related-work concern (Tu et al.'s split-set idea) was also addressed with a differentiation: Tu et al. is a cryptographic/indistinguishability framework requiring multiple model trainings across splits, while this paper's "split" is within the target subset and requires no extra training. What may still be outstanding for UFX4 is largely a positioning issue: the reviewer asked for clearer articulation of why the metric is "better" and what properties an evaluation metric should satisfy (robustness/consistency), and the rebuttal answers this in prose and experiments, but the paper may still lack a formal "properties" framing that would fully satisfy this critique.

Reviewer 3Ap9's concerns were addressed to the point that the reviewer explicitly maintained the positive rating.

Reviewer QQ14's concerns were also largely addressed. The "metric gaming" point is acknowledged as a general problem, and the rebuttal argues that dependence in latent representations is harder to directly game than accuracy or distributional metrics, while also positioning SDE as a practical complement when retraining or training attacker models is infeasible.

**Reviewer Scores:**

Reviewer kGt2 was already close to the acceptance boundary. Their main blockers were the lack of a clean theoretical sanity check and missing cost analysis, both of which were partially addressed in the rebuttal. While they might still view the deep-network theory as approximate, I expect they would have nudged their score up slightly to 6, or at least remain the same score 4.

Reviewer UFX4 gave a reject score, driven largely by concerns about unclear motivation and seemingly ad-hoc design choices. The rebuttal directly addressed the design-choice issues and clarified how the method differs from cryptographic split-set evaluations, but the reviewer's concern about formal properties and framing is only partially resolved. I would not expect a full reversal, but with the added explanations and empirical validations, it is plausible the score would move from 2 to 4.

Reviewer 3Ap9 was strongly positive and explicitly stated that their concerns were addressed and that they maintained their rating.

Reviewer QQ14 was already supportive and I would expect no numerical change.

---

### Decision · Program_Chairs · 2026-01-26

Accept (Poster)